# Stray and Owner-Relinquished Cats in Australia—Estimation of Numbers Entering Municipal Pounds, Shelters and Rescue Groups and Their Outcomes

**DOI:** 10.3390/ani13111771

**Published:** 2023-05-26

**Authors:** Diana Chua, Jacquie Rand, John Morton

**Affiliations:** 1School of Veterinary Science, The University of Queensland, Gatton, QLD 4343, Australia or jacquie@petwelfare.org.au (J.R.); johnmorton.jemora@gmail.com (J.M.); 2Australian Pet Welfare Foundation, Kenmore, QLD 4069, Australia; 3Jemora Pty Ltd., P.O. Box 5010, Geelong, VIC 3219, Australia

**Keywords:** cats, surrendered, stray, euthanized, pound, shelter, Australia, reclaim, rehome, rescue, semi-owned, transfer

## Abstract

**Simple Summary:**

Lack of comprehensive and accurate data on the frequency of stray and owner-relinquished cats entering municipal pounds, animal welfare shelters, and rescue groups across the states and territories in Australia is an impediment to targeting management of domestic cats to where it is most needed. Our study aimed to evaluate availability of data as well as collect and analyze comprehensive data to establish a baseline to measure future improvements in domestic cat management. Data were collected for 2018–2019 by email and phone where not publicly available. Unavailable municipal pound data were imputed based on known data and the human population. We estimated a total of 179,615 stray and surrendered cat admissions to pounds, shelters, and rescue groups in Australia in 2018–2019 (7.2/1000 human residents) and that 5% of admissions were reclaimed, 65% rehomed, and 28% euthanized. Municipal councils operating their own pounds rehomed 26% and euthanized 46% of cat intake compared to 65% rehomed and 25% euthanized for welfare organizations. Data collation and analyses at a national level would be facilitated by open public access to standardized intake and outcome data for municipal pounds, shelters, and rescue groups. This would highlight where improvements are most needed and serve as a baseline to track the impact of new policies, protocols, and legislation. More effective management of domestic cats will ultimately benefit the community, cats, and wildlife.

**Abstract:**

Access to comprehensive municipal pound, animal welfare shelters, and rescue group data for admissions and outcomes for stray and owner-relinquished cats in Australia is currently lacking. This hinders effective assessment of existing management strategies for domestic cats by animal management agencies. Our study aimed to estimate the numbers of cat admissions and intakes to Australian municipal council pounds, animal welfare organizations (excluding smaller animal welfare organizations thought to have annual cat intakes of less than 500), and animal rescue groups and their respective outcomes for 2018–2019 (pre-COVID). Unavailable municipal council data were imputed based on known data and council human populations. Only Victoria and New South Wales had publicly available municipal data, and only RSPCA had publicly available data in all states. We estimated a total of 179,615 (7.2/1000 human residents) admissions to pounds, shelters, and rescue groups in 2018–2019, with an estimated 5% reclaimed, 65% rehomed, and 28% euthanized. Reclaim rates were low across all the agencies. Councils operating their own pound had nearly double the euthanasia rate (estimated at 46%) compared to animal welfare organizations (25%). Rescue groups rehomed an estimated 35% of the total number of cats rehomed by all agencies. The upper quartiles of councils with intakes of >50 cats in Victoria and New South Wales had estimated euthanasia rates from 73% to 98%, and 67% to 100%, respectively. We recommend that comprehensive municipal pound, shelter, and rescue statistics be routinely calculated using standardized methods and made available publicly in a timely fashion. This would inform management strategies to optimize live outcomes and therefore reduce the negative mental health impacts on staff tasked with euthanizing healthy and treatable cats and kittens.

## 1. Introduction

Free-roaming cats in urban and peri-urban areas pose a significant problem in Australia and many countries in the world [1,2]. Commonly cited complaints include nuisance behaviors such as fighting and soiling, potential wildlife predation, and zoonotic risk to humans and other animals [3,4,5]. Management of these cats poses an ongoing challenge and is associated with high cat intake and euthanasia in facilities operated by local governments (municipal council pounds) and animal welfare organizations [6]. This is psychologically detrimental to shelter and pound staff who are often tasked with killing large numbers of healthy and treatable cats and kittens [7,8]. In addition, cat management exerts financial pressure on local governments and animal welfare organizations. Increasingly, a One Welfare approach is being advocated, which aims to balance and optimize the wellbeing of animals, people, and their physical and social environment [9,10,11].

In Australia, municipal councils (local government organizations) are responsible for urban cat management in all states and territories, except South Australia, Tasmania, and Northern Territory. However, not all councils operate shelters (called pounds) for cats, and of those that do, some outsource this role to animal welfare organizations. Thus, many welfare organizations are a hybrid model and have one or more contracts to accept either all cats, or just those that are not returned to owners after the mandatory holding period. These organizations also accept strays (lost, injured, and unowned) and owner-relinquished cats directly from the public [12]. Both municipal pounds and shelters operated by welfare organizations are physical establishments. In contrast, animal rescue groups typically operate on a home-based foster care system without a physical shelter and limit admissions to capacity [13]. They usually accept strays and owner-relinquished cats, and many accept transfers in from council pounds and animal shelters [14].

In Australia, the classification of cats varies according to individual state government legislation. However, the classification recommended by Australia’s leading national welfare agency, the Royal Society for Prevention of Cruelty to Animals (RSPCA) [15] and used by the Commonwealth Government [16] and some state governments [17,18] has been used here, with cats categorized based on how and where they live. Feral cats live and reproduce in the wild (e.g., forests, woodlands, grasslands, deserts) and survive by hunting or scavenging; none of their needs are fulfilled intentionally by humans. Domestic cats are owned, semi-owned (fed intentionally by humans), and unowned (obtain food from humans unintentionally) living in and around cities, towns, and rural properties. Depending on the state, owned cats can belong to an individual, household, business, or corporation, and most or all their needs are supplied by their owners. 

Semi-owned cats are fed or cared for to a certain extent by people who do not acknowledge ownership, are of varying sociability, and may be associated with one or more households [1,19,20,21]. Unowned cats are indirectly dependent on humans but may have casual and temporary interactions with humans and are of varying sociability, including some that are unsocialized. They may live in groups (colonies) in urban or peri-urban environments and obtain food from humans unintentionally, including around rubbish tips, food outlets, and fishing harbors [15]. Both semi-owned and unowned cats likely originate from litters born to free-living semi-owned and unowned cats, as well as lost, abandoned, or free-roaming owned cats. Free-roaming domestic cats may be a source of complaints, resulting in them being trapped and impounded, or brought to a shelter by a concerned member of the public. 

Based on these definitions, feral cats are not found in urban or peri-urban areas and are not typically represented in shelter admissions to municipal council pounds and animal welfare facilities. However, some domestic cats are inappropriately classified as “feral” based on behavior on admission, or shortly after, and are usually euthanized within 24 h [22]. In some councils, kittens are also euthanized if the queen is deemed “feral”.

Unidentified owned, semi-owned, and unowned cats are typically classified as stray cats in shelter data and represent 80–100% of admissions to municipal council pounds and 60–80% of admissions to animal welfare agencies (higher for those with local government contracts to provide municipal pound services). The remainder of these admissions are largely owner-relinquished cats [6,22].

### 1.1. Statistics from Australia

After entry to a municipal pound or animal welfare shelter, cats that are not returned to their owner are either rehomed/sold, transferred into another organization, euthanized, or have another outcome (e.g., lost, stolen, died). Euthanasia of healthy and treatable cats as a population control strategy is a source of contention in the community [1,23]. Although the current euthanasia proportions for cats in Australia are unknown, a study of cat intakes to Victorian municipal councils in 2016–2017 reported that of outcomes occurring in that year, on average, 48% of cats were euthanized, compared to 8% of dogs, and the highest quartile of councils euthanized between 67% and 98% of cats. Intake per capita was reported as 7/1000 residents, and numbers euthanized as approximately 3/1000 residents [6]. For 13% of outcomes, the cat was returned to the owner [6]. In contrast, the largest animal welfare organization in Australia (RSPCA) from the corresponding period reported outcomes nationally, and 23% of cats were euthanized (versus 13% for dogs), and only 5% of cats were returned to owner [24].

### 1.2. Statistics from USA and UK

In comparison, it was estimated that 3.2 million cats (9.7/1000 residents) were admitted to shelters in the USA operated by welfare organizations, municipal governments, and rescue groups, with 17% of cats euthanized annually (1.6/1000 residents), based on collated statistics in 2019 [25,26,27]. In a 2010 study in the United Kingdom, 156,826 cats (2.5 per 1000 residents) [28] were admitted to animal welfare organizations and rescue groups, with 13% (0.2/1000 cats) euthanized [2,29]. The municipal councils in the United Kingdom have no legal responsibility to take in stray cats [30]. 

### 1.3. Knowledge Gap and Aims

The social and economic ramifications of utilizing euthanasia as a population control strategy have since fueled critical appraisal of the management strategies in municipal and animal welfare shelters in many countries, particularly for urban and peri-urban stray cats. A key limitation to our understanding of the scale of the problem in Australia is that there are no existing national or state-based systems for the surveillance of numbers of admissions of stray and owner-relinquished cats or their outcomes. This gap in meaningful data hinders the evaluation of existing animal management strategies, including affirmation of effective approaches and identification of inadequacies. It also does not facilitate tracking of improvements over time [31]. Access to and knowledge of such detailed data could drive more efficient allocation of resources in municipalities and animal welfare organizations [32] and potentially improve the outcomes for stray and owner-relinquished cats, as well as the mental and psychological well-being of staff in shelters and pounds. 

The aims of this study were, firstly, to evaluate the accessibility of data sources and to develop a methodology to accurately estimate of the numbers of cat admissions to Australian municipal council pounds, animal welfare organizations, and animal rescue groups for 2018–2019 (pre-COVID), or the most current year of available data, and to report the outcomes of those admissions on a national and state level and by type of organization. Secondly, this study aimed to examine whether intakes and outcomes for municipal councils were associated with either the council’s human resident population or whether the council was urban or not. Thirdly, based on the findings of the study, we aimed to make recommendations for implementation of strategies aligned with a One Welfare philosophy, which will benefit animals, people, and the environment, and reduce the number of healthy and treatable cats killed in pounds and shelters. 

Our hypotheses were, firstly, that improvements in the proportion of cats returned to owners, rehomed, or euthanized have occurred since previous peer-reviewed state-based [6] or organizational reports [22,33,34]; secondly, that live outcomes were higher for animal welfare organizations than municipal councils operating their own pound; thirdly, that urban councils would have lower per capita intakes and better live outcomes than non-urban councils; and, fourthly, that there are a number of life-saving programs yet to be embraced in Australia for various reasons, including legislative barriers.

## 2. Materials and Methods

### 2.1. Organizations

All councils were identified through websites and contacted via email and telephone to determine if they managed stray and owner-relinquished cats. Animal welfare organizations in each state that operated physical animal shelters were identified through online sources and local knowledge to identify those with estimated annual intakes of 500 or more cats. Smaller animal welfare organizations thought to have annual cat intakes of less than 500 were excluded from the study due to logistical difficulty in identifying and contacting them. PetRescue, a national organization supporting animal rescue groups in Australia, provided most of the rescue group data across Australia [35]. Data for two additional New South Wales rescue groups not represented in the PetRescue data were provided by Geoff Davidson from Justice4Max [36]. Large rescue groups not represented in either of these sources, but known to the authors, were contacted by email/telephone. 

### 2.2. Definitions of Admission, Intake and Outcomes

For the purpose of this study, we defined admission and intake differently. We defined an admission as commencing when a cat entered a municipal council pound, animal welfare shelter, or animal rescue group from the “outside world” (i.e., not from another organization) and ending only when one of four outcomes occurred: returned to owner, rehomed (i.e., allocated to a new owner), other outcome (e.g., died in the shelter, stolen, escaped etc.), or euthanized. Thus, admissions do not commence with a transfer in from another organization. During an admission, the cat might be transferred out to another organization (i.e., (another) municipal council, animal welfare organization, or animal rescue group) with the admission then ending with one of the four outcomes in a different organization from where the admission commenced.

In contrast, we defined intake to a particular organization as all entries to that organization: (a) owner-relinquished or stray cats brought in by the general public or authorities, etc., and (b) cats transferred in from other organizations. For example, if a cat was relinquished to a municipal council pound, then transferred out to a rescue group from which it was subsequently rehomed, the cat would contribute one admission but two intakes—one each to the municipal council and rescue group. Furthermore, if that cat was subsequently impounded or relinquished again to the same municipal council or a different organization, that would constitute both a further intake and a second admission at the state and national levels. Thus, our definition of intake was as inferred in industry-standard definitions [37]. Those definitions are organization-based and so do not include the concept of admissions, which, when there are transfers, apply across organizations. Thus, intake and admission are distinct concepts, and both are useful for different reasons. Intakes reflect the workload for each organization, whereas admissions reflect the experience from the cat’s perspective from entry to an organization to either leaving the care of any organization or euthanasia. It is important that at the state and national level, admissions are used rather than the sum of intakes, because the latter results in double counting of animals transferred from one organization to another. These concepts are depicted in Appendix A: Figure A1.

We assumed transfers out of one organization and into another occurred only within (and not between) states and territories. If there were no transfers within a state/territory, each intake would also be an admission. When transfers occur, the sum of intakes for all organizations within the state/territory exceeds the number of admissions for the same year for that state/territory. Numbers for states/territories and nationally are appropriately described using admissions whereas statistics for municipal councils, welfare organizations and animal rescue groups are appropriately described using intakes.

Unincorporated territories in New South Wales, Victoria, South Australia, and Northern Territory were parts of remote Australia that have not been “incorporated into the area of a local government” and lack “accountable local representation” [38]. Unincorporated territories were unlikely to operate municipal pounds and were excluded in our study. Australian Capital Territory is also an unincorporated territory with no local government or council-operated pound for cats, and cats were instead accepted by the RSPCA and rescue groups and reported in their data. 

### 2.3. Data Collection

Intake and outcome data for cats and numbers of cats in care that were in the possession of the pound or shelter (including cats in foster care) at the beginning and end of the year were sought for the Australian financial year 1 July 2018–30 June 2019 (i.e., pre-COVID), and if not available, for the most recent calendar or financial year where data were available. 

For most major animal welfare organizations in all states, and the municipal councils in Victoria and New South Wales, pertinent data were available online. Where unavailable, attempts were made to obtain data through direct correspondence (email and/or telephone contact) with the organization or via Right to Information applications. Sources of data for welfare organizations and rescue groups other than those listed with PetRescue or the NSW government are summarized in Appendix A: Table A1.

All municipal councils in South Australia, Queensland, Northern Territory, Western Australia, and Tasmania were sent an email detailing the study and requesting relevant data. Repeated follow-up emails and/or telephone calls were made to non-responders. For New South Wales, the Department of Local Government [39] published municipal council data on their website. For Victoria, their municipal council data were mostly available publicly via their Domestic Animal Management Plan on their council websites, and where available for some councils, 2018–2019 data were obtained from annual reports. Where unavailable, more recent data (2020–2021) were used. If required, follow-up telephone calls or emails were conducted to clarify data. For the Australian Capital Territory, there was only one state-run pound (a facility to impound seized or roaming animals) operated by Domestic Animal Services (government agency), and they did not accept cats.

#### Data Requested from Municipal Councils

For each municipal council, we sent a questionnaire via email to request their pound/shelter statistics and determine the following information:If the council took in and managed reclaiming (return to owner) and/or rehoming of cats, using its own facility/facilities (i.e., managed one or more pounds for cats);Agencies into which any cats were transferred.

Where a council utilized a service provider (usually an animal welfare organization) either to manage its council pound, or to receive all cats or only cats not returned to owners (transferred out of council care), we sought to determine if the council data were included in the relevant animal welfare organization’s statistics. If this was unclear, we assumed that council data had been included in the welfare organization’s reported data. This same assumption was applied in cases where data were not available for councils with a contractual obligation with a welfare organization, such as the 84Y agreement in Victoria [40]. These 84Y agreements (under the Domestic Animals Act 1994) are legal agreements between councils and shelters, veterinary clinics or rescue groups for the “seizure, holding and disposal of dogs and cats” [40].

While some municipal councils provided complete data as requested, others provided incomplete or no data. To minimize missing data, these were obtained through the “Right to Information” process from six municipal councils with human populations greater than 50,000 in Queensland (Brisbane, Bundaberg, Cairns, Logan, Rockhampton, and Townsville), as well as for two Victorian municipal councils (Wyndham and Geelong). “Right To Information” is a legislative process to “make more information available to members of the community and provide a framework for the lawful management and handling of individuals’ personal information”, and they incur administrative fees [41]. 

### 2.4. Estimation of Unavailable Council Data

By state/territory, for municipal councils with cat pounds and complete available data, we calculated the total pooled intake over 12 months as a ratio of the total human population for those councils, and we excluded councils with no pounds or cat management and those with pound services operated by an animal welfare agency. Using that calculated cat intake per 1000 human residents of those councils with available data, we then imputed the missing intake data for each council known to operate a pound, but whose data were not available. For each of these councils, the imputed intake was calculated using that council’s human resident population multiplied by intake per 1000 residents for councils with available data pooled. The same process was repeated to impute numbers of each outcome type (return to owner (RTO), rehomed, transferred out, had another outcome, or euthanized), and as well as numbers in care at the start and end of the year. Human populations used for each municipal council and for each state or territory were those on 30 June 2018 [42]. Specific details for imputing intake and outcome numbers for Western Australia and South Australia are shown in Appendix A: Table A2.

### 2.5. Animal Rescue Groups 

Centralized reporting was not mandatory for animal rescue groups in any state or territory other than New South Wales. Access to comprehensive rescue group data was further complicated by the large number and variety of types of animal rescue groups in every state, making this process a logistic challenge. PetRescue, a national welfare organization advertising animals for adoption on behalf of municipal councils, welfare agencies, and animal rescue groups, provided most of the relevant rescue group data for all states. A total of 41,355 cats were recorded with PetRescue or from NSW data as being available for adoption through 416 animal rescue groups across Australia in 2018–2019, and we used these numbers by state/territory as estimated intakes for that year. 

For PetRescue data in 2018–2019, of cats of known sources (69%), 41% were strays (recorded as community cats), 25% owner-relinquished, 25% transferred in from council pounds, and 9% transferred in from animal welfare shelters. Of the of 41,261 cats from the PetRescue website, 93% were recorded as having been rehomed, and 7% of cats were recorded as having been removed from the PetRescue website, and their outcomes would have included being adopted by the foster carer, transferred out to another organization, lost/escaped, died, or euthanized. For our calculations, we assumed that 2% of cats entering rescue groups were euthanized [36]. 

In New South Wales, animal rescue groups that were registered as Rehoming Organizations with the government were exempt from registration requirements for animals in their custody for the first 12 months [43], provided they submitted annual reports to the New South Wales Office of Local Government. Data for 2017–18 were available from an animal advocacy group, Justice4Max [36], through an informal request via the GIPA (Government Information Public Access) Act), which is a legislative platform to request access to government information [44]. However, many of these submitted data were for dogs and cats combined, although it was estimated that one quarter of the combined intake were cats [36]. Furthermore, the 2018–2019 data were less comprehensive than previous years (pers. Comm. Geoff Davidson, Justice4Max). All the cat-specific rescue groups included in Justice4Max data were also included in Pet Rescue data, with the exception of Cat Defence Network and Tikki Animal Rescue. Data for these two groups were therefore added to the numbers of cats admitted by rescue groups recorded by PetRescue. Some very large rescue organizations were also not included in the New South Wales Office of Local Government data as they were not registered Rehoming Organizations. They were ineligible for registration exemptions because they also rehomed animals directly from the public and not just from council pounds. These organizations were contacted directly but did not respond with data on request. 

### 2.6. Calculation of Intake, Admission and Outcome Statistics

Discrepancies between numbers for intake and outcomes for municipal councils and animal welfare organizations are shown as positive or negative gaps (Appendix A: Table A3). Discrepancies for admissions at the state and national level were also shown in a similar way. Cats in foster care were typically counted as part of the shelter population unless formally transferred out to a rescue group. Discrepancies between the sum of the number of cats transferred in from municipal councils and welfare agencies and intakes obtained for animal rescue groups were calculated as shown in (Appendix A: Table A3).

#### 2.6.1. Total Numbers of Admissions

Although we used summary data from each organization rather than records for individual animals, it was possible to estimate numbers of admissions for the year by state/territory. Assuming that transfers out and in occurred only among agencies within the same state or territory, and that transfers in the study year were for admissions that commenced in the study year, the total number of admissions for each state or territory were calculated as the sum of intakes (i.e., including numbers transferred in) minus the sum of numbers transferred out across the municipal councils and animal welfare organizations within the state or territory (rescue groups were assumed not to transfer cats out).

#### 2.6.2. Outcomes

Numbers of outcome events occurring in the study year were available. These were for cats that were in care at the start of the year or entered during the study year and had an outcome during the study year, but not for cats in care at the end of the year (as these had no outcome until the following year). Estimated numbers of each outcome event (returned to owner, rehomed, transferred out, euthanized, had another outcome) were expressed as numbers/1000 human residents. Our data were not suitable for directly calculating percentages of admissions that ended in, variously, the cat being returned to owner, rehomed, euthanized, and having another outcome (the appropriate measures from an animal welfare and ethics perspective) as only aggregated data were available from each organization (rather than individual records for each intake) and transferred cats could not be tracked across organizations. However, as surrogate estimates of these, we calculated percentages of the admissions that ended in the study year for these outcomes. For intakes, we also calculated percentages of the total number of outcome events in the study year for these outcomes. Example calculations are shown in Appendix A: Figure A1.

In addition, we calculated percentages of outcomes where the outcome was rehomed and euthanized, respectively using as the denominator the total number of outcomes less the number where the outcome was “returned to owner” (RTO). This facilitates comparison of rehoming performance among agencies with differing success returning cats to owners (determined in part by the proportion of admissions that are “strays” that have the potential to be returned to owner versus owner-relinquished cats, for which RTO is not a possible outcome). 

### 2.7. Association between Human Resident Population or Urban Status and Intake per 1000 Residents and Outcomes

To examine whether intakes for municipal councils were associated with either the council’s human resident population or whether the council was urban or not, we used municipal council classifications from the Australian Classification of Local Governments to identify which councils were urban (including regional city) and which were rural [45]. Relationships between annual council intake per 1000 residents and each of council human resident population and urban status (city or other) were assessed based on councils where intakes were available (i.e., not imputed) and the human resident population was at least 5000, using linear regression. The individual council was the unit of analysis. Council intake per 1000 residents was simultaneously regressed on council urban status (urban or rural) and resident population using generalized linear models with Gaussian error distribution and log link, fitted using the -glm- command in Stata (version 17, StataCorp LCC, College Station, TX, USA). Separate models were fitted for each of New South Wales and Victoria, the only 2 states with at least 18 eligible councils of each urban status. Relationships between council urban status (urban or rural) and resident population with percentage of unclaimed outcomes where (a) the cat was rehomed and (b) where the cat was euthanized were assessed in a similar way but using generalized linear models with binomial error distribution and logit link.

## 3. Results

### 3.1. Availability of Data and Contributions of Imputed Data

There were 348 possible sources of data from local municipal government agencies (councils) and animal welfare shelters: 305 councils (excluding those identified as not impounding cats) and 43 animal welfare organizations thought to have intakes of ≥500 cats per year in their state or territory in each of the six states and two territories. We excluded two animal welfare organizations in Queensland that operated in 2019 but closed during the COVID-19 pandemic and did not reopen. Of the 348 possible sources, we obtained data for 207 (59%), comprising 185 (60%) of the 305 municipal councils and 22 (52%) of the 43 welfare organizations. Rescue group data were provided by PetRescue for 414 rescue groups that advertised cats with them in 2018–2019, and data for 2 additional rescues were sourced from the NSW Office of Local Government data via Justice4Max (total 416), but the total number of rescue groups in Australia was unknown.

Table 1 shows the total number of councils in each state and territory as well as the proportions of councils where data were available for our study, of those known or assumed to operate a pound (not managed by a welfare agency). The Australian Capital Territory’s (ACT) territorial government body did not manage cats (there is no local government body in ACT), and no council in Tasmania appeared to operate cat impoundment facilities.

Victoria had the highest proportion of councils whose data were publicly available. Of all 79 Victorian councils, 77 published data online in their Domestic Animal Management Plans (DAMP). Right to Information was used to obtain data for the two councils with missing data (human populations >200,000). No imputation was done for Australian Capital Territory (ACT). The proportions of estimated total number of admissions for the state or territory that consisted of imputed intakes for councils assumed to operate their own pounds and impounded cats were: New South Wales: 0.3%, Victoria: 1%, Northern Territory: 4%, Queensland: 5%, South Australia: 7%, and Western Australia: 26%, and 5% when pooled nationally. 

Of the animal welfare organizations known or thought to have annual intakes of 500 or more, RSPCA had comprehensive, publicly available data at the state/territory and national levels. In Victoria, Animal Aid, Lost Dogs Home, Cat Protection Society, and Geelong Animal Welfare Society had data available online as did Ten Lives (Tasmania). Some organizations thought to have intakes of ≥500 per year did not have publicly available data, but some provided them on request for the study (Table A1). 

### 3.2. National Results

The national total number of admissions to municipal councils, animal welfare organizations, and animal rescue groups in 2018–2019 was estimated to be 179,615 or 7.2 admissions/1000 human residents (Table 2). In total, 176,965 admissions ended in 2018–2019 and of these, for 5% (0.4/1000 residents), the cat was returned to owner, for 65% (4.6/1000 residents) the cat was rehomed, and for 28% (2.0/1000 residents), the cat was euthanized (50,022 cats). 

Intakes by municipal councils, animal welfare organizations, and animal rescue groups combined totaled 192,584, of which municipal councils accounted for 30% (57,917), animal welfare organizations accounted for 48% (93,312), and animal rescue groups accounted for 21% (41,355; Table 2). Of the 41,355 intakes of cats by 416 rescue groups, 28,753 (70%) were admissions directly from the general public, accounting for 15% of the estimated total national intake. 

For New South Wales, the estimated number transferred out of municipal council pounds and animal welfare organizations (6547) exceeded the total intake (6453) to animal rescue groups reported by PetRescue plus numbers from Cat Defence Network and Tikki Animal Rescue, even though (a) most of these 6547 transfers were probably to animal rescue groups and (b) animal rescue groups would also have received intake from the general public. This discrepancy was probably largely because not all rescue groups utilized PetRescue to advertise cats available for rehoming. Some larger rescue groups in NSW including Mini Kitty Commune, were not registered with PetRescue or with the NSW Office of Local Government, and although they were directly approached, declined to provide data for the study. Thus, for New South Wales, we took animal rescue groups pooled intake as 6547, knowing this was almost certainly an underestimate. (We assumed 98% of these were rehomed and 2% euthanized as in other states and territories.)

The main animal welfare organization in Australia, RSPCA, had an intake of 47,388, which was 51% of the intake to animal welfare organizations (93,312), 35% of the intake to animal welfare organizations and animal rescue groups combined (134,667), and 25% of the total national intake (192,584). The percentages of outcome events where the cat was returned to owner were similar for both councils (7% or 4007/56,283) and welfare organizations (6% or 5125/92,296). Rescue groups were assumed to have no animals returned to owner. The percentage of outcome events where the cat was rehomed for welfare organizations (65%) was more than double that for councils (26%). Animal rescue groups were assumed to have rehomed 98% of their intake (Max4Justice/NSW). The percentage of outcome events where the cat was transferred out was higher for municipal councils (20%) than for welfare organizations (2%). It was assumed that the rescue groups did not transfer cats out. For councils pooled across Australia, the percentage of outcomes where the cat was euthanized was 46%, which was close to double that for welfare organizations (25%). It was assumed that rescue groups euthanized only 2% of their intake. 

### 3.3. Comparisons between States/Territories

For admissions, South Australia ranked the highest on a per capita basis (10.9/1000 residents) and New South Wales was lowest (4.9/1000 residents) (Figure 1; Table 2). Victoria had the highest total number of admissions (52,893). 

Percentages of outcome events where the cats were returned to owner were generally low across all states and territories, varying from 2% to 7%, with highest percentages in Victoria, Australian Capital Territory, and Queensland (Figure 2; Table 2). 

Tasmania had the highest numbers rehomed per capita (7.3/1000 residents or 84% of all outcomes) and New South Wales had the lowest (2.7/1000 residents; 49% of outcomes) (Figure 3; Table 2).

Australian Capital Territory had the lowest proportion of cats euthanized (13% of outcomes in the year) while Northern Territory euthanized the highest proportion (32%). On a per capita basis, the Australian Capital Territory also had the lowest number of cats euthanized (0.8 cats/1000 residents) and South Australia had the highest (3.1 cats/1000 residents). Both Victoria and New South Wales were comparable in having the highest total number of cats euthanized (14,664 and 14,464 cats, respectively), double that of the state with the next highest number (Queensland, 7801) (Figure 4; Table 2).

### 3.4. Comparisons within Organizations between States/Territories

#### 3.4.1. Municipal Councils

Based on total pooled intakes (including imputed intakes) for municipal councils that operated pound services, councils in Western Australia had the highest estimated intake per 1000 residents (3.7/1000; Figure 5; Table 3). In comparing outcomes for municipal councils, South Australia and Northern Territory had the lowest RTO percentage of 3%, while Queensland had the highest at 11% (Table 3). South Australian councils were estimated as rehoming the highest percentage (48%), while Northern Territory and Queensland rehomed the lowest at 0% and 12%, respectively, and had high euthanasia percentages (85% and 51%, respectively; Table 3). 

The lowest euthanasia percentages for councils at the state/territory level were South Australia (44%), Victoria (43%), and New South Wales (41%) with the latter two having the most council data available (rather than imputed) (Figure 5; Table 3). Municipal council estimates were most robust for New South Wales and Victoria. For other states, numbers were imputed for substantial proportions of their municipal councils, so caution is warranted when interpretating estimates for those states, especially for Western Australia and Northern Territory.

There was substantial variation between councils, and for New South Wales and Victoria where comprehensive data were available, the highest quartile of municipal councils with intakes of 50 or more cats had euthanasia percentages between 67% and 100%, and 73% and 98%, respectively (Table 4). Of councils with populations over 200,000, the highest euthanasia percentages were for Hume (Victoria, population 224,423), which euthanized 1626 out of 1962 cats (83%), and Logan (Queensland, population 345,098 residents), which euthanized 1996 out of 2474 cats (81%).

#### 3.4.2. RSPCA

Comparing RSPCA results by state/territory, RSPCA New South Wales had the highest intake of 14,676 (equating to 1.8/1000 residents) while RSPCA Northern Territory had the lowest (387 or 1.6/1000 residents) (Figure 6; Table 3). RSPCA RTO percentages ranged from a low of 2% in Northern Territory to highs of 7% in both Queensland and Victoria and 8% in Australian Capital Territory. In the latter territory, the municipal council did not impound cats; stray and owner-relinquished cats were mainly accepted by RSPCA (pers. comm. Domestic Animal Services/ACT). Percentages where the cat was rehomed varied from 53% in New South Wales to 91% in Northern Territory. RSPCA New South Wales had the highest euthanasia percentage (40%) while RSPCA Northern Territory had the lowest percentage euthanized (6%), followed by Queensland with 13% euthanized (Figure 6; Table 3).

#### 3.4.3. Other Welfare Organizations

Other welfare organizations (i.e., not including RSPCA or animal rescue groups) in all the states and territories pooled had a low RTO percentage of 5%, but percentage rehomed was 65% (Figure 7; Table 2). These were virtually identical to RSPCA pooled nationally with an RTO percentage of 6% and rehoming percentage of 66% (Figure 7). By state/territory, the other welfare organizations with the highest rehoming percentage were: New South Wales—Cat Protection Society (98%), Tasmania—Just Cats (90%), Western Australia—Cat Haven (83%), Victoria—Cat Protection Society (84%), and Queensland—Animal Welfare League in (80%) (Table 3). 

Of the individual animal welfare organizations, Lost Dogs Home in Victoria euthanized the highest percentage (48%) followed by Animal Welfare League South Australia (46%). The Cat Protection Society in New South Wales had the lowest percentage euthanized (1%) with Lort Smith Animal Hospital in Victoria having the next lowest percentage (5%) (Table 3). In comparison to the largest welfare organization, RSPCA, with a percentage euthanized of 25% nationally, the percentage euthanized by other welfare organizations was the same at 25%. (Figure 7; Table 2).

#### 3.4.4. Rescue Groups

Animal rescue groups intake directly from the general public accounted for 15% (28,753/192,788) of the national total intake, and based on our assumption that they rehomed 98% of their total intake, they accounted for 35% (40,528/115,630) of all the cats rehomed nationally.

### 3.5. Trends over Time

In New South Wales where comprehensive municipal council pound data were available over multiple years, intakes from 2016–2017 to 2018–2019 [39] were utilized to establish trends in intake and numbers euthanized over these 3 years (Table 5). Corresponding data were available for RSPCA and Animal Welfare League in New South Wales. For municipal councils, over the three years, annual intakes increased with approximately corresponding proportionate changes in numbers euthanized. Annual intakes for RSPCA declined slightly but with proportionately larger decreases in numbers euthanized, and annual intakes for Animal Welfare League declined considerably from 2016–2017 to 2018–2019, with a corresponding proportionate decline in number euthanized.

### 3.6. Associations between Each of Council Urban Status and Human Resident Population with Intake, and Percentages of Unclaimed Outcomes That Were (a) Rehomed and (b) Euthanized

For New South Wales councils, intakes per 1000 human residents decreased markedly with increases in human resident population (changing by an estimated factor of 0.8 after adjustment for urban status; 95% CI 0.7 to 0.9; *p* < 0.001) for each additional 1000 population) and were higher for urban councils than rural councils (estimate adjusted for human resident population: 6.8 times higher after; 95% CI 3.2 to 14.5; *p* < 0.001). These relationships were not evident for Victorian councils (estimated proportional change per additional 1000 population: 1.00; 95% CI 0.99 to 1.00; *p* = 0.499; estimated ratio for urban relative to rural 0.79; 95% CI 0.44 to 1.41; *p* = 0.424). 

There was also no compelling evidence for relationships between either council urban status and human resident population and percentage of unclaimed outcomes that were rehomed for New South Wales (adjusted estimated odds ratio per additional 1000 population: 1.003; 95% CI 0.997 to 1.009; *p* = 0.334; estimated odds ratio for urban relative to rural 4.42; 95% CI 0.94 to 20.91; *p* = 0.061) or Victoria (adjusted estimated odds ratio per additional 1000 population: 1.000; 95% CI 0.986 to 1.015; *p* = 0.979; estimated odds ratio for urban relative to rural 3.08; 95% CI 0.58 to 16.5; *p* = 0.188).

Similarly, there was no compelling evidence for relationships between either council urban status and human resident population and percentage of unclaimed outcomes that were euthanized for New South Wales (adjusted estimated odds ratio per additional 1000 population: 1.000; 95% CI 0.994 to 1.005; *p* = 0.869; estimated odds ratio for urban relative to rural 0.50; 95% CI 0.20 to 1.27; *p* = 0.146) or Victoria (adjusted estimated odds ratio per additional 1000 population: 0.998; 95% CI 0.983 to 1.013; *p* = 0.759; estimated odds ratio for urban relative to rural 0.41; 95% CI 0.08 to 2.17; *p* = 0.296).

For each outcome variable, within each state, *p*-values for interaction between urban status and human resident population were high (≥0.252), so no interactions were assumed. However, estimated coefficients for interaction terms were imprecise. Thus, our hypothesis that urban councils would have lower per capita intakes and better live outcomes than non-urban councils was not supported when adjusted for population.

## 4. Discussion

### 4.1. National Admissions and Intakes

In this study, we estimated intake of stray and owner-relinquished cats to municipal pounds, animal welfare shelters, and animal rescue groups in 2018–2019, and their subsequent outcomes, as well as admissions and outcomes at the state and national level. This was based on available data collated from the different agencies, and we imputed municipal data where data were not available. The estimated total number of cat admissions nationally was 179,615 or 7.2/1000 residents in 2018–2019. The estimated total intake nationally (where cats entering one organization and transferred out to another contribute an intake for each organization) was 192,584 or 7.7/1000 residents. This was less than the estimated total intake of 9.7 cats/1000 residents for municipal and animal welfare shelters and rescue groups in United States in 2019 [46]. 

In United Kingdom, the only available national data were collated in 2010, where the total intake into participating shelters and rescue groups was 2.5/1000 residents [28]. This was less than half the intake for animal welfare organizations and animal rescue groups combined in our study (5.4/1000 residents). However, the UK data were based on a 39% response rate from the estimated number of shelters and rescue groups, and is therefore an underestimation. RSPCA in United Kingdom reported a total intake of 0.5/1000 residents in 2019 [47], compared to 1.9/1000 residents for Australian RSPCAs pooled from our study. In contrast to dogs, municipal councils and local authorities have no legal responsibility to deal with stray cats in the United Kingdom [30], and the proportion of animals transferred from one organization to another was very low. Therefore, the UK intake is more appropriately compared with our national admissions of 7.2/1000 residents. In addition to underreporting, several factors likely contribute to differences in per capita intake in the United Kingdom compared to Australia and the United States. These include lack of municipal facilities accepting stray cats, and that people finding a stray cat are advised by the RSPCA [47] that they cannot respond to calls about healthy stray cats, and that the finder can rehome a stray cat if they cannot find an owner. This is reflected in the lower proportion of stray to owner-relinquished cats in the United Kingdom data compared to that reported for Australia [22,29,33].

### 4.2. Admissions by State/Territory

Admission rates on a per capita basis ranged from a low of 4.9/1000 residents in New South Wales to a high of 10.9/1000 residents in South Australia (Table 2). In Australia, domestic animal management is legislated at the state level, and local municipalities can enact by-laws that are more stringent than the state laws, but not more lenient. For example, registration (licensing) of cats is required in half the states and territories (New South Wales, Victoria, Western Australia, and South Australia in 2018). In the other states/territories, (Queensland, Tasmania, Northern Territory, Australian Capital Territory), registration is not required. It will be required in Australian Capital Territory from 2022, and local governments in the other states can enact by-laws requiring registration, although most do not currently require registration. Based on our data showing that states requiring registration of cats had the highest (South Australia) and lowest (New South Wales) numbers of admissions per capita, it is clear that additional or other factors were influencing admission rates per capita. 

Differences in numbers of cat admissions per capita between states/territories, in part, likely reflect differences in state legislative requirements for local governments to manage cats, and the response of local governments to this obligation. In Victoria (8.2/1000 residents) and Queensland (7.4/1000 residents), local governments are required to manage cats under their respective Acts, and most either provide cat traps and/or actively trap cats for residents in response to calls about found or nuisance cats. For example, under the Victorian Domestic Animals Act 1994, “found or stray cats in the possession of a person, other than the owner, must be handed over to the local council as soon as possible” [48]. In South Australia (10.9 cats/1000 residents), 83% of municipal councils did not actively manage domestic cats in their localities. Instead, most councils hired cat traps out to members of the public, with instructions to transport trapped cats to welfare organizations, veterinary clinics, or another council with an impound facility (S. Hazel, personal communication (15 December 2020)). This is reflected in the high intake per 1000 residents to South Australia RSPCA and Animal Welfare League. Given the minimal costs to councils to hire traps but not impound cats, there was little incentive for them to implement strategies to decrease intake, such as targeted sterilization programs.

In Northern Territory, which had the second lowest admission rate of 5.5 cats/1000 residents, there is no state based legislation for management of urban cats [49]. Furthermore, Northern Territory has a considerable number of indigenous communities which do not operate municipal facilities or actively impound cats (30% of the Northern Territory population is indigenous, compared to 13% of national population).

### 4.3. Intakes by Agency

Animal welfare organizations accounted for nearly half of national admissions (48%), compared to municipal councils (30%) and rescue groups (15% directly from general public) (Table 2). Of the animal welfare agencies, RSPCAs accounted for 25% of the national admissions and 51% of intake into welfare agencies. Although the animal welfare agencies are mainly publicly funded, with some state government funding, they were also often contracted by the municipal councils to provide pound services for local governments, commonly seen in Victoria (RSPCA and Lost Dogs Home) and Queensland (RSPCA and Animal Welfare League). In South Australia, intake for the animal welfare organizations was about four times higher per 1000 residents than for councils in that state, an indication that most municipal councils did not impound cats, but instead hire cat traps and direct the public to take cats to the main animal welfare agencies (RSPCA and Animal Welfare League), (Table 3). The proportion euthanized (37%) by welfare agencies in South Australia was also the highest of all states. In contrast to other states where welfare agencies with council contracts receive funding for providing these services, animal welfare organizations in South Australia often received minimal funding. The cost estimated by Animal Welfare League South Australia for rehoming a cat is over AUD$700, and costs for a 30-to-40-day length of stay may exceed AUD$1000–1500 per cat [22]. Prioritizing resources to reduce intake would have the dual benefit of reducing the number of cats euthanized and providing cost savings, which could then be channeled into broadening sterilization and pet retention programs to further reduce intake and rehoming programs to increase the live release rate [50,51].

### 4.4. Strategies to Reduce Intakes and Admission Rates

Most cats entering shelters have no owner identification and are classified as “strays”. In a 2015 study, only 9% of stray cats entering RSPCA Queensland were microchipped [52]. Unidentified free-roaming owned cats and semi-owned cats likely comprise the majority of stray cats, and 60% of people bringing a stray cat to RSPCA shelters said they had been feeding it for more than a month [20]. Most semi-owners feed one to two cats at private residences, but multiple cats (often called colonies) also congregate at private residences where there is a food source, and other sites such as aged care facilities, social housing, industrial complexes, and universities [53,54]. Stray cats are brought to shelters and pounds by members of the public because they believe the cats will be better off in the shelter [20] or because they are causing a nuisance. The majority of cats entering shelters and pounds are less than 6 months old, highlighting the need for sterilization programs [33]. Highest intakes are from low socioeconomic areas [19,33,55], where the cost of sterilization and microchipping is often unaffordable, as is cat-proof fencing. At the time of the study, weekly income for 20% of Australian households (average of 2.4 people) was less than AUD$650, with higher proportions of low-income households in socioeconomically disadvantaged areas [42]. Recent Australian research demonstrated that high-intensity sterilization programs predominantly for owned and semi-owned cats, targeted to areas of high cat admissions, or high call-volume related to found or nuisance cats, were effective in rapidly reducing cat intake and euthanasia from the target area [56,57]. For example, sterilizing 7 to 33 cats/1000 residents a year reduced intake by 30% to 50% from the target areas within 12 to 24 months [56,57,58]. 

Although owner-relinquished cats constitute a minority of admissions, strategies to assist owners to keep their cats, for example, by providing free advice on behavioral problems, assisting with costs of veterinary treatment, including sterilization, and licensing (registration) costs effectively reduce the incidence of pet relinquishment [22,51]. It is also recommended that animal welfare agencies and municipal councils encourage “home to home” adoptions of cats by directing people wishing to relinquish their cats to relevant websites, such as PetRescue’s Home2Home site [59]. Lack of pet-friendly rental accommodation has been identified as a major factor for relinquishment of adult cats [34]. Recently, both Victoria and Queensland legislated that landlords will no longer be allowed to refuse pets in rental properties without a reason deemed valid by the state government, and in New South Wales, pet ownership cannot be reasonably refused, but only for individual owners of units or townhouses in strata titled properties. 

### 4.5. National Reclaim Rate 

The national RTO rate was only 5% (0.4/1000 residents) (Table 2). This was similar to the estimated RTO rate of 2% to less than 5% in the United States [26,27,37]. In comparison, the RTO rates for dogs in both Australia (48%) [60] and United States (23%) [27] are much higher than for cats.

### 4.6. Reclaim by State/Territory

Victoria had the highest RTO rates of 7%, while Northern Territory had the lowest (2%) (Table 2). Northern Territory was the only state or territory without mandatory microchipping laws. Victoria was the first state in Australia to implement state microchipping legislation, in 2005, with Queensland following in 2008. Victoria had the highest number of cats per 1000 residents that were newly microchipped and registered annually on the largest national microchip registry database (Central Animal Records) from 2008 to 2016 [61].

### 4.7. Reclaim by Agency

RTO rates were similar at the national level for municipal councils (7%) and welfare organizations (6%), (Figure 7; Table 2). The highest mean RTO rates for councils operating their own pound were in Queensland (11%) and Victoria (10%), and the best performing municipal councils from these states both had RTO rates of 24%. However, these means were less than previously reported (13%) from most (70/79) Victorian councils, and the upper quartile of these councils achieved RTO rates of 17% to 60%. These data are not directly comparable with our data, because only 38 of these councils operated their own pound, and most had contracts with welfare agencies to take all cats, or only those not returned to owners [6]. 

The generally low RTO rates across both municipal pounds and welfare organizations reflect the fact that most cats entering shelters have no identification. Although classified as strays in shelter records, most were likely unidentified outdoor, owned and semi-owned cats that were not reclaimed because the owners and carers did not search for them at the shelter or pound within the mandated holding period (usually 3–8 days for an unidentified cat), perhaps in the belief that the cat would return of its own accord [62,63,64]. 

### 4.8. Strategies to Increase Reclaim Rates

Semi-ownership of cats is not uncommon in Australia, with 3% to 9% of self-selected survey respondents regularly feeding an average of 1.5 cats they do not consider they own [3,21]. When free sterilization, microchipping, and registration are provided for these cats, most semi-owners feeding one to two stray cats will take full ownership of the cats they are feeding with their details registered on the cat’s microchip and registration databases, successfully transforming most semi-owned stray cats to owned, sterilized, and identified cats [56,57,58]. Strategies to provide free or highly subsidized sterilization and microchipping programs in low socio-economic areas for both owned and semi-owned cats will decrease cat intake and likely increase RTO rates, and in turn reduce the number of cats euthanized. These programs should be embraced as a core strategy by municipal councils and welfare organizations to manage urban cats. In New South Wales, the RTO rates for municipal councils (4%) were less than half those of Queensland (11%) and Victoria (10%) (Table 2). In New South Wales, the AUD$59 (2022) life-time registration fee for cats, when added to the cost of microchipping and sterilizing, is an additional cost barrier for low-income owners and for semi-owners adopting the cats they are caring for. Anecdotally, some cats are not being microchipped at the time of sterilization to avoid payment of registration fees, because the New South Wales state microchip registry can be used to identify owners who have not registered their cat. The costs associated with urban cat management for councils (estimated at AUD$9.825 m for 20,053 cats at an average cost of AUD$490 per impounded cat (per comm J. Verrinder AWL Qld, August 2021) exceed by approximately seven times the income derived from cat registration fees in New South Wales (<AUD$1.4 m, and not considering costs to collect fees; per comm Office of Local Government), and it is recommended that registration for cats be removed to reduce barriers to sterilization and microchipping. Of note, Queensland abolished registration fees for cats in 2013, just 5 years after they were introduced, to remove the regulatory burden on local governments [65]. Although local governments in Queensland can enact local laws requiring registration, most of the larger local governments have abolished it [65]. Since collecting data for this study, New South Wales has introduced a permit requirement for cats not sterilized by 4 months of age ($81), even if the cat was not acquired by 4 months of age. The total fee of AUD$140 for the permit and registration would pose a further barrier to semi-owners adopting the cats they are caring for and should also be abolished. 

In 37% of stray cats with a microchip (9%), the contact details were not current [52]. Incorrect contact data substantially affected the proportion of cats returned to owner (41% versus 75% of microchipped cats with no data issues). Strategies to facilitate updating microchip details by owners are recommended. For example, email reminders with a link to the database company website have been shown to increase the frequency at which owners check and update their contact details for their pet [61]. Other strategies welfare agencies and local governments could use to increase RTO rates include short message service (SMS) reminders, providing free cat collars with owner’s contact details for cats being adopted, or visiting veterinarians to facilitate RTO by members of the public or neighbors [62,63,66]. Although Check a Chip events are successful for dogs, they are more problematic for cats because of challenges with transport and the safe containment of cats at such events [52]. Private veterinary clinics could play a more active role in checking and updating owners’ details [62,67,68]. 

### 4.9. National Rehome Rates and by State and Agency

Across Australia, 65% of cats were rehomed (Table 2). This is similar to the estimated annual rehoming percentage of 66% in United States in 2019 [26] and 77% in United Kingdom in a 2010 survey [28]. At the national level, welfare agencies rehomed 65% while municipal pounds rehomed 26% of their intake (Figure 7). Municipal pounds also transferred out a further 20% mainly to rescue groups for rehoming (Table 2).

### 4.10. Strategies to Increase Rehoming Rate

Data collected by PetPoint in USA over the past eight years indicate an increase in adoptions in the last decade, and this has become a key driver, alongside decreasing intake, for the recent decline in euthanasia across the United States [69,70]. RSPCA Qld achieved a large increase in rehoming rates for cats from 34% in 2011 to 74% in 2016 [22]. This was accomplished by innovative paid advertising to increase shelter adoptions and partnerships with retail businesses for offsite adoptions. To increase the availability of adoptable cats and kittens, they doubled the number of cats within the foster network from 2011 to 2016. The foster care system benefits older or timid and shy cats that are at increased risk of euthanasia, and kittens requiring bottle feeding [22,71]. Adult and elderly cats are less likely to be adopted compared to kittens [72], and the length of stay in a shelter can result in inhibited, defensive, and disruptive behaviors due to stress [72,73]. Strategies to fast-track adoptions, simplifying the adoption process, off-site adoptions including allowing foster carers to adopt directly from their home, rather than returning the cat to the shelter for the adoption process, coupled with adoption promotions (one-day adoption events), innovative advertisements (e.g., “Geek Chic” digital campaign), and collaborations with retail establishments to function as adoption centers, can help move eligible cats into appropriate homes more efficiently [22]. 

### 4.11. Role of Rescue Groups in Reducing Numbers Euthanized

Rescue groups rehomed 35% of all cats rehomed in Australia, resulting in a very substantial impact on decreasing numbers euthanized, particularly in municipal facilities. Rescue groups play a key role in optimizing live release rates of stray and owner-relinquished cats in pounds and shelters, by reducing their presence in pounds and shelters and enhancing their access to potential adopters [22,74]. Rescue groups operate on a foster care model, which provides a conducive environment for socialization, enrichment, and behavior conditioning, and their euthanasia rate was assumed to be 2% [74]. 

### 4.12. Euthanasia—National, by State and Agency 

Nationally, 2.0 cats per 1000 residents were euthanized (Table 2), which was similar to United States in 2019 (1.6/1000 residents) [26] and considerably higher than for United Kingdom in 2010 (0.2 per 1000 residents), but only 39% of organizations were represented [29]. Of all outcomes nationally in the current study, 28% of admissions were euthanized. This statistic was not reported in those sources for United States and United Kingdom, although in the United States, euthanasia percentage (17%) was expressed as a proportion of adjusted intake, which excluded transfers in, and in the UK (13%), transfers in were a very small proportion of intake, so our data are broadly comparable between countries. 

Northern Territory and New South Wales had the highest euthanasia percentages at 41% and 32%, respectively (Table 2). Both Australian Capital Territory and Tasmania had the lowest euthanasia percentage of 13%. This is largely attributed to the minimal active cat management by the municipal councils, with the welfare organizations (RSPCA Australian Capital Territory and RSPCA Tasmania, Just Cats and Ten Lives in Tasmania) playing a prominent role in managing the stray and owner-relinquished cats via council contracts. It is important to note that Western Australia’s euthanasia results should be interpreted with caution as data were imputed for 78% of their councils that were thought to manage cats, and it is possible that those not reporting had higher euthanasia rates. 

Euthanasia percentages for municipal council-operated pounds for the states/territories ranged from 41% (New South Wales), 51% (Queensland), 53% (Western Australia), to 85% (Northern Territory), and the proportions of unclaimed cats that were euthanized were even higher (Table 3). Municipal councils traditionally respond to calls relating to nuisance cats by providing a trap cage or actively trapping the offending cats and euthanizing large numbers of healthy cats. For a well-resourced country such as Australia, this is socially unacceptable and is not consistent with a One Welfare philosophy of enhancing the welfare of animals, people, and the environment [9,10,11]. 

Return to field (RTF) programs (also called shelter neuter return, SNR) are another highly effective way to manage healthy stray cats admitted to the shelters and reduce euthanasia, particularly for cats that are less socialized and at risk of euthanasia or a long length of stay before they are adopted [75,76]. These cats are sterilized and then returned to where they were found. This is an effective way to ensure that semi-owned and free-roaming owned cats find their way home. As our data show, most cats are not returned to owners from the shelter, although the hold period is typically a minimum of 3 days. Other studies have shown that most owners do not start looking for a missing cat until 3 days have passed [63,77]. In the United States in 2022, 5.6% of cats were returned to field, which represented 12% of stray cat intake [37,78]. However, this is illegal in all states of Australia under various legislation relating to abandonment, containment to property, and biosecurity [15]. Relocating poorly socialized cats to be sterilized working cats for rodent control around factories, farms, barns, and other businesses is an additional method of achieving live outcomes for cats that are otherwise at high risk of euthanasia, but it takes more time and resources than RTF [79,80].

In the United States, due to inherent and systemic bias, animal control policies are over-enforced in low-income communities, resulting in worse health and welfare outcomes for disadvantaged people and their pets [50,69]. This is also likely the situation in Australia, given the higher per capita intake rates in low socioeconomic areas [19]. For carers of stray cats, lethal enforcement-centered management of the cats they are providing care for also has deleterious effects on their psychological health and quality of life and is not aligned with a One Welfare philosophy [81]. A paradigm shift from the traditional punishment-orientated approach to support-based models of animal control (support for owners and semi-owners to provide care for their cat) addresses complaints and aligns the animal welfare field with the modern human social justice movement [50]. This strongly aligns with the Pets for Life concept where owners and carers are supported to care for their cat (versus relinquishment), which is a cost-effective strategy to implement [51]. Pets for Life is “driven by social justice and guided by the philosophy that a deep connection with pets transcends socioeconomic, racial, and geographic boundaries, and no one should be denied the opportunity to experience the benefits, joy, and comfort that come from the human–animal bond” [51].

Apart from the heavy financial burden placed on the community from impounding and euthanizing cats, there is also an implicit human cost associated with the prevalent use of euthanasia as a population control strategy in municipal pounds and shelters. Often, an underestimated number of pound or shelter workers grapple with post-traumatic stress associated directly or indirectly with the euthanasia of animals in their care [7,8,82,83] culminating in substance abuse, cardiovascular conditions, suicidal inclinations [23,82,84], and high staff turnover. There is a hence growing emphasis on the training of shelter staff in applying coping strategies associated with the stress of euthanasia [23]. The increased recruitment and training of shelter staff further adds to the current sheltering costs, both financially and socio-emotionally. Animal welfare agencies are increasingly focusing on strategies to reduce intake, and to not take in more animals than they can provide care for (capacity for care) [75,85,86].

Intake rates into municipal councils and animal welfare agencies are an important consideration in the efforts to reduce the number of cats euthanized. There is a close correlation between the numbers of cats admitted and numbers euthanized [87]. Given that 50% to 75% of cats entering shelters and pounds were born in the preceding 6 months, and most emanate from low socioeconomic suburbs [19,33,55], affordable sterilization programs are urgently needed and can be a cost-effective way to address this [57]. Targeted high-intensity, free sterilization programs for owned, semi-owned, and unowned cats are rapidly effective at reducing cat intake into the shelters and pounds, and correspondingly, decreasing the numbers euthanized. Two Australian studies recently reported decreases in the numbers of cats euthanized by 45% to 90% in 12 to 24 months following implementation [56,57,58]. These programs are highly successful at getting semi-owners to take ownership of the cat/s they are feeding, cats which are often timid and shy, and hence at a higher risk of euthanasia in a shelter or pound [58].

Trap neuter and return (TNR) [88] is another strategy utilized in other countries and effectively reduces shelter and pound intake and euthanasia, when targeted and high-intensity [76,89,90]. It is practiced on a small scale in Australia [53,54] and is effective, but is illegal in all states/territories under various state and local government legislation relating to biosecurity (feeding feral animals), animal care and protection (abandonment), and domestic animal management (wandering cats). It is strongly recommended that legislative changes occur to enable the community, municipal governments, and animal welfare organizations to benefit from access to TNR and RTF as management tools for free-roaming urban cats to improve animal and human welfare.

## 5. Limitations of the Study

### 5.1. Quality and Availability of Data

This study highlighted significant challenges in accessing and collating meaningful municipal council and welfare agency data. At the time of writing, there is still no centralized database collating pound and shelter data across Australia, nor are there standard definitions and methods for generating statistics from individual agencies. In the United States, although no mandated centralized reporting system exists, national estimates are tabulated through the Shelter Animals Count (SAC) and other known sources of estimates [26]. The SAC is their most recent effort to collate national shelter data and is dependent on voluntary reporting of statistics and of the 3200 organizations that have reported, 14% are municipal councils, 36% welfare agency shelters, and 50% rescue organizations [37], and the collated data had been essential in providing “insights into national sheltering trends” [69]. In 2019, based on voluntarily reported data from SAC, adjusted intake (total intake minus transfers) was 6.5 cats/1000 residents and total intake (including 14.2% of transfers in) was 7.5 cats/1000 residents [37,78]. In contrast, adjusted intake estimated by the American Society for Prevention of Cruelty to Animals (ASPCA) was 9.7 cats/1000 residents, suggesting that 33% of adjusted intake are not reported in SAC data [25,26,27,37,78].

#### 5.1.1. Municipal Councils

New South Wales and Victoria had the most comprehensive municipal council data available and were therefore the most transparent, while Queensland and Western Australia were least transparent with data available for less than 30% of municipal councils thought to manage cats. The majority of municipal councils that were approached to provide data chose not to respond to requests for data or required “Right to Information” application for release of pound data (each RTI application cost AUD$52.60 in Queensland and AUD$41.25 in Tasmania as of July 2021).

New South Wales was the only state that mandated the collation and publication of municipal council statistics online annually [39]. In Victoria, most municipal councils reported intake and outcome data for their impounded dogs and cats under their Domestic Animal Management Plans (DAMP) published online, which is reviewed every 4 years, but most DAMPs were not updated annually. As of January 2023, all shelter and pound establishments in Victoria must collect and report data related to animal intake and their outcome events to Animal Welfare Victoria for publication on their website for each calendar year [91]. It is recommended that other states also implement this requirement, so resources can be better directed to areas they are most needed to reduce intake and euthanasia.

Data collected should include intake and sources of intake because strategies to reduce stray cat intake are potentially different from those aimed at helping owners retain their pets rather than relinquishing them. The number of juveniles under 6 months of age should also be reported to assess the effectiveness of sterilization programs. Outcomes reported should include numbers returned to owner, rehomed, euthanized, transferred (and organization transferred to), and other (e.g., lost, stolen, escaped). It is recommended that they are reported as total number and cats per 1000 residents, because when municipal councils are very active in reducing intake, the proportion of intake euthanized may cease to be a valid comparison of performance. It is also recommended that length of stay (LOS) be reported because it is increasingly recognized as a critical factor in shelter management, with implications for animal health, well-being, sheltering costs, and ultimately a shelter’s capacity to save lives [92]. Multiple studies have identified LOS as the most significant risk factor for illness in shelter dogs and cats in the United States. With illness comes the need for treatment, reduced welfare, and a yet more prolonged stay [93,94,95]

#### 5.1.2. Welfare Agencies

Of the main welfare organizations operating in more than one state, only the RSPCA was transparent providing annual data that were available to the public. The state Animal Welfare League organizations, except for Queensland, did not have data available for the public, but all provided it on request for the study. Most of the other smaller state-based welfare agencies did not respond to requests to provide data or were unknown to us (Table A1). Therefore, our study likely underestimated the number of cats admitted and euthanized at the state and national level because we were unable to impute missing welfare agency data on a per capita basis, as was done for municipal councils.

#### 5.1.3. Rescue Groups

Our direct requests to rescue groups that did not have data listed with PetRescue or the NSW Office of Local Government were largely unsuccessful in obtaining data, and because the total number of rescue groups in Australia was unknown, we were unable to impute missing data, resulting in an underestimation of cats received directly from the general public by rescue groups.

### 5.2. Lack of Transparency

The difficulties encountered in data collection for this study highlight the ongoing lack of transparency and, therefore, accountability of many public and private agencies. Transparency of performance is important for identifying agencies and geographic locations with high cat intakes or poor outcomes for cats, and hence for the staff involved. In 2018–2019, the Lost Dogs Home (Victoria) had a cat intake of 10,549 and euthanasia rate of 48%, and therefore clearly had not implemented practices to improve outcomes for cats and staff, as compared to the similar sized Cat Haven (Western Australia) whose intake was just under 9000 cats with a euthanasia rate of 11%. At the time of writing, the Lost Dogs Home euthanasia rate was 39% [96]. At both the organizational and state level, it is important that high performers can be identified and acknowledged, and their policies and procedures shared with those that have poorer outcomes. Donors and rate payers should have greater ability to compare municipal councils, welfare agencies, and rescue groups, given that they are supporting these organizations voluntarily as donors or involuntarily as rate payers. Potentially, this might lead to greater incentives for agencies to improve their performance.

Knowledge and identification of what practices save lives [97], particularly over the last 10–15 years, are currently being shared at national [98] and international conferences [99,100,101]. For example, assessing suitability of cats for rehoming based on behavior over a minimum of three days rather than one day, leads to a much smaller proportion being deemed “feral” and euthanized. This could substantially lower numbers of healthy cats killed [22]. Facilitating the comparison of agency performance for board members and donors may lead to better prioritization of resources to improve processes shown to increase the live release rate of stray and owner-relinquished cats [58]. Reducing intake of cats is important to reduce the numbers killed, and recent data from Australia confirm findings from the United States that high intensity sterilization programs targeted to locations of highest cat intake are effective in reducing intake and euthanasia [53,54,56,57,58,76,89,90,97,102,103,104].

Our data were not suitable for directly calculating percentages of intake/admissions that ended variously in the cat being RTO; rehomed; transferred out; euthanized; and having another outcome (the appropriate measures from an animal welfare and ethics perspective) as only aggregated data were available from each organization (rather than individual records for each intake). However, the percentages of outcome events are probably close to the appropriate measures. Relative to the appropriate method, for intakes where the cat entered late in the study year, the percentages of outcome events are probably biased upwards for events that occur soon after entry (e.g., returned to owner, euthanized). However, we also included outcomes for cats in care at the start of the study year, and for these, the percentages of outcome events are probably biased upwards for events that occur longer after entry (rehomed). The magnitudes of these biases would be only modest, if for most cats, the outcome occurs soon after entry. An RSPCA Queensland study reported that 92% of cats had an outcome by 90 days, with RTO and euthanasia occurring largely within the first 7 to 14 days, while average length of stay was 30 days [22].

Most of the admissions in our study were likely unique cats based on the results of a study of 195,387 cats entering RSPCA shelters nationally over a 4-year period, which found 98% of cats had only a single entry [33]. Multiple entries by the same cat were detected based on microchip data because all cats were microchipped by the shelter before being rehomed and most cats being reclaimed were microchipped. Most cats had only two entries, but up to seven entries were recorded.

## 6. Conclusions

It is strongly recommended that consistent standards in recording, aggregating, and reporting of data in municipal councils, animal welfare organizations, and rescue groups statistics are established across the Australian states and territories, and the aggregated statistics made publicly available online for all individual agencies. These data should include source (impounded by local government, strays from general public, owner relinquishments, transferred in from other organization), age (adult cat versus kitten), outcomes (RTO, rehomed, transferred, euthanized, other), as well as length of stay. It is recommended that state databases be created and updated annually as has been legislated in Victoria. These should be generated using nationally agreed standard definitions and methods for generating statistics from individual agencies. This in turn would facilitate evaluation of existing management strategies and their impact on numbers of cats admitted and euthanized. Being able to identify agencies and geographic locations with high cat intakes or poor outcomes for cats would inform targeted interventions and better allocation of resources for management strategies known to be effective. Reducing the number of free-roaming cats and the numbers of healthy and treatable cats and kittens killed would positively impact the mental health of shelter and municipal pound staff and is aligned with the One Welfare philosophy. This is particularly critical given our results show that municipal council intake and euthanized numbers are increasing over time.

In summary, the results of our study set the benchmark to compare over time the impact of animal management policies and practices on agency performance and on the stray and owner-relinquished cat population entering municipal facilities, animal welfare shelters, and rescue groups.

## Figures and Tables

**Figure 1 animals-13-01771-f001:**
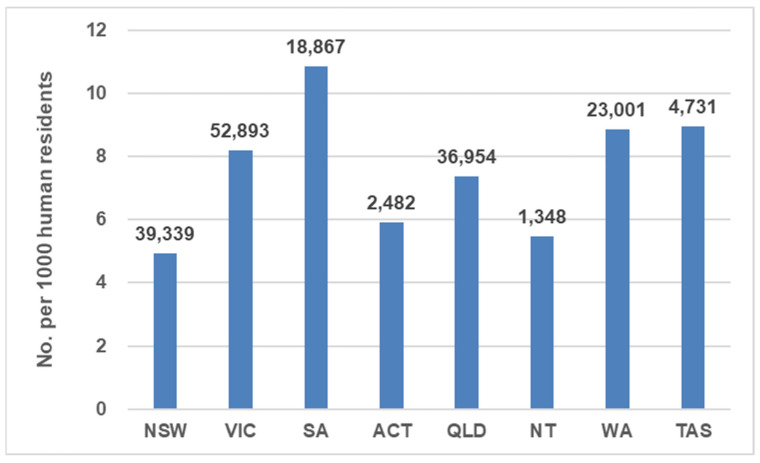
Numbers of cat admissions to municipal councils, welfare organizations and animal rescue groups per 1000 human residents and total number by state/territory in 2018–2019. Abbreviations: NSW (New South Wales); VIC (Victoria); SA (South Australia); ACT (Australian Capital Territory); QLD (Queensland); NT (Northern Territory); WA (Western Australia); TAS (Tasmania).

**Figure 2 animals-13-01771-f002:**
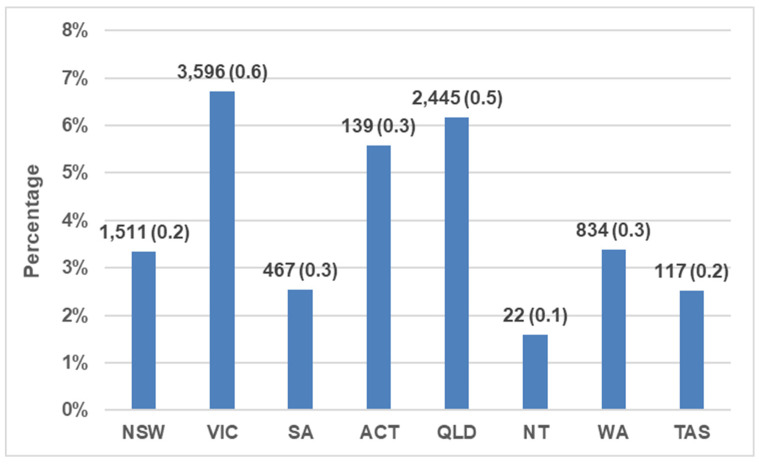
Percentages of outcome events during the year where the cat was returned to owner, total number returned, and in brackets, number returned to owner per 1000 residents in 2018–2019.

**Figure 3 animals-13-01771-f003:**
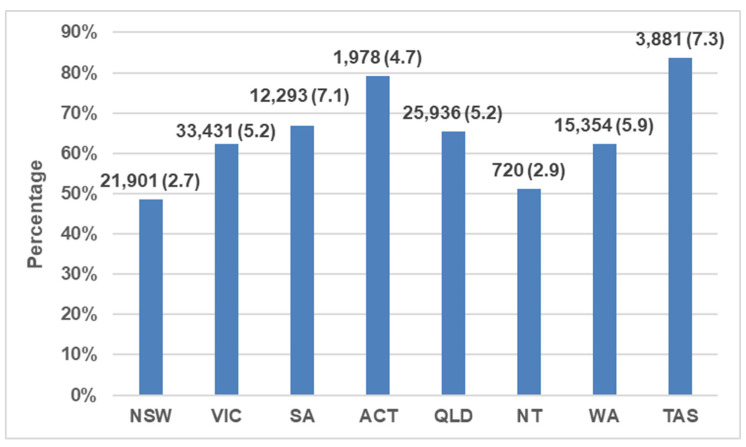
Percentages of outcome events in the year where the cat was rehomed, total number of rehomes, and in brackets, number rehomed per 1000 residents in 2018–2019 by state/territory.

**Figure 4 animals-13-01771-f004:**
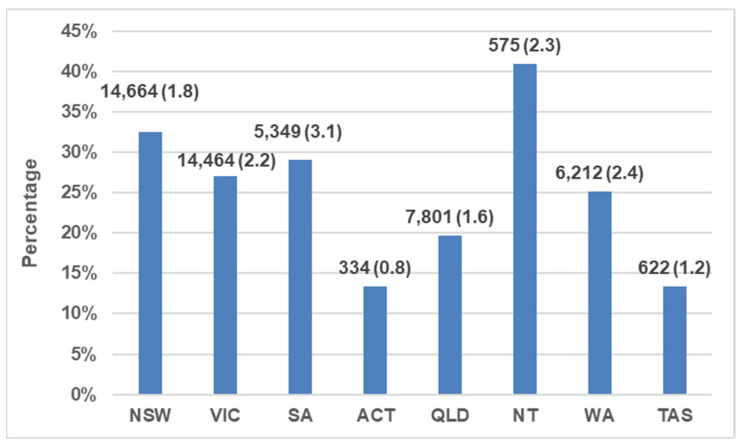
Percentages of outcome events during the year where the cat was euthanized, total number euthanized, and in brackets, number euthanized per 1000 residents in 2018–2019 by state/territory.

**Figure 5 animals-13-01771-f005:**
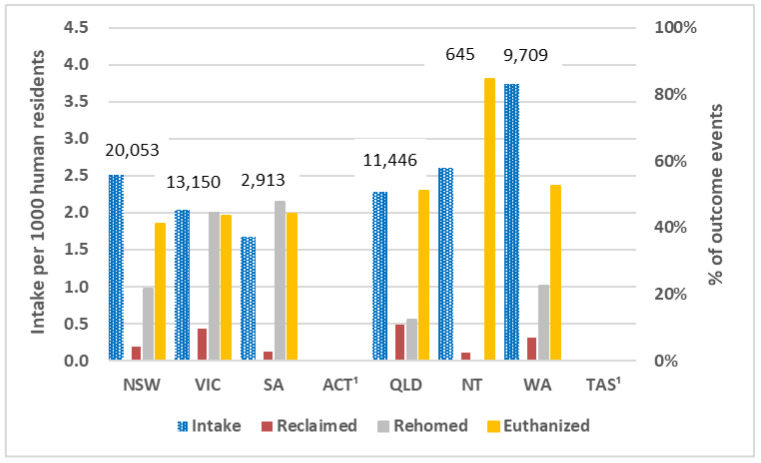
Intakes for municipal councils that operated pounds (numbers above bars), intakes per 1000 human residents (left-hand axis) and percentages of outcome events where the cat was reclaimed, rehomed, and euthanized (right-hand axis) in 2018–2019 by state/territory. ^1^ The state government in ACT and the municipal councils in TAS did not operate a pound for cats.

**Figure 6 animals-13-01771-f006:**
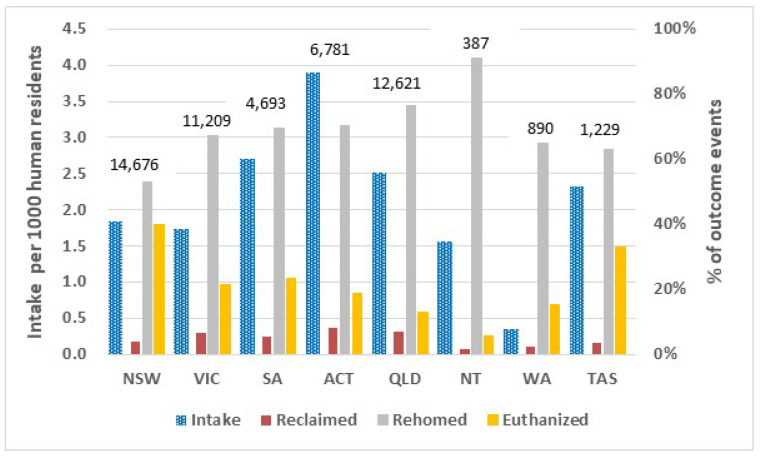
Intakes for RSPCA (numbers above bars), intakes per 1000 human residents (left-hand axis), and percentages of outcome events where the cat was returned to owner, rehomed, and euthanized (right-hand axis) in 2018–2019 by state/territory. Abbreviations: NSW (New South Wales); VIC (Victoria); SA (South Australia); ACT (Australian Capital Territory); QLD (Queensland); NT (Northern Territory); WA (Western Australia); TAS (Tasmania).

**Figure 7 animals-13-01771-f007:**
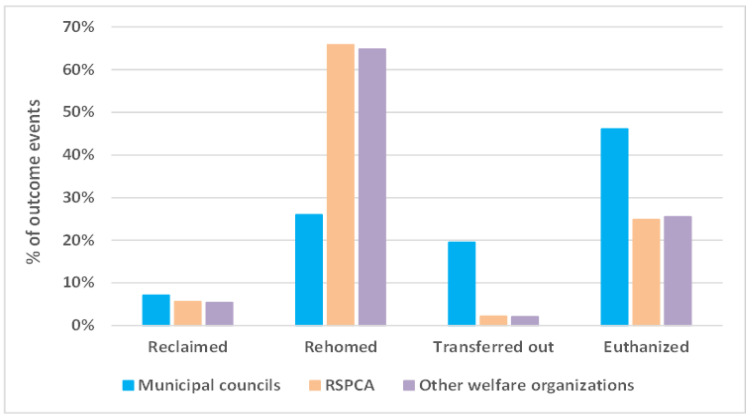
Percentages of outcome events where the cat was returned to owner, rehomed, transferred out, and euthanized in 2018–2019 nationally for municipal councils that operated pound services, RSPCA, and other welfare organizations (i.e., not including RSPCA or animal rescue groups).

**Table 1 animals-13-01771-t001:** Total numbers of councils in state, number and proportion operating own pound, and number and proportion supplying data, including data obtained by Right to Information (RTI).

State	Total Number of Municipal Councils	Number of Municipal Councils Operating Their Own Pound	Proportion of Municipal Councils Operating Their Own Pound	Number Operating Their Own Pound That Supplied Data	Proportion Operating Their Own Pound That Supplied Data
New South Wales	128	111	87%	107	96%
Victoria	79	38	48%	37 (1/38 by RTI)	97%
South Australia	69	12	17%	4	33%
Queensland	78	60	77%	18 (6/18 by RTI)	30%
Northern Territory	17	3	18%	1	33%
Western Australia	149	82	55%	18	22%
Tasmania	29	0	0%	N.A.	N.A.
Australian total	549	306	56%	185	60%

**Table 2 animals-13-01771-t002:** State/territory totals and national admissions/intakes and outcomes for cats entering municipal council pounds, welfare organizations, and animal rescue groups in 2018–19.

State/Territory (Human Population)	Admissions/Intake for Year		Outcomes during Year								Discrepancy ^5^
			Reclaimed	Rehomed		Transferred Out	Released Alive ^3^	Euthanized		Other Outcomes ^4^	No. (%) ^6^
	No. ^1^	No. (Per 1000 Residents) [%] ^2^	No. (Per 1000 Residents) [%] ^2^	No. (Per 1000 Residents) [%] ^2^	% Rehomed of Unclaimed	No. (Per 1000 Residents) [%] ^2^	No. (Per 1000 Residents) [%] ^2^	No. (Per 1000 Residents) [%] ^2^	% Euthanized of Unclaimed	No. (Per 1000 Residents) [%] ^2^	
**Admissions**:											
New South Wales (7,980,168)	39,339	4.9	1511 (0.2) [3%]	21,901 (2.7) [49%]	50%		29,960 (3.8) [66%]	14,664 (1.8) [32%]	34%	513 (0.1) [1%]	+88 (+0%)
Victoria (6,462,019)	52,893	8.2	3596 (0.6) [7%]	33,431 (5.2) [62%]	67%		38,183 (5.9) [71%]	14,464 (2.2) [27%]	29%	909 (0.1) [2%]	+387 (+1%)
South Australia (1,736,527)	18,867	10.9	467 (0.3) [3%]	12,293 (7.1) [67%]	68%		12,829 (7.4) [70%]	5349 (3.1) [29%]	30%	235 (0.1) [1%]	+601 (+3%)
Australian Capital Territory (420,379)	2482	5.9	139 (0.3) [6%]	1978 (4.7) [79%]	84%		2126 (5.1) [85%]	334 (0.8) [13%]	14%	34 (0.1) [1%]	0 (0%)
Queensland (5,009,424)	36,954	7.4	2445 (0.5) [6%]	25,936 (5.2) [65%]	70%		31,226 (6.2) [79%]	7801 (1.6) [20%]	21%	453 (0.1) [1%]	+330 (+1%)
Northern Territory (247,058)	1348	5.5	22 (0.1) [2%]	720 (2.9) [51%]	52%		434 (1.8) [31%]	575 (2.3) [41%]	42%	6 (0.0) [0%]	0 (0%)
Western Australia (2,594,181)	23,001	8.9	834 (0.3) [3%]	15,354 (5.9) [62%]	64%		8402 (229.8) [34%]	6212 (2.4) [25%]	26%	148 (0.1) [1%]	+496 (+2%)
Tasmania (528,298)	4731	9.0	117 (0.2) [3%]	3881 (7.3) [84%]	86%		3017 (5.7) [65%]	622 (1.2) [13%]	14%	21 (0.0) [0%]	0 (0%)
National (24,982,688)	179,615	7.2	9132 (0.4) [5%]	115,494 (4.6) [65%]	69%		124,625 (5.0) [70%]	50,022 (2.0) [28%]	30%	2319 (0.1) [1%]	+1902 (+1%)
**Admission outcomes:**											
Admissions nationally	176,965		9132 [5%]	115,494 [65%]				50,022 [28%]		2319 [1%]	
**Intakes**:											
Municipal councils	58,121 *	2.3	4007 (0.2) [7%]	14,775 (0.6) [26%]	28%	11,004 (0.44) [19%]	29,785 (1.2) [53%]	26,036 (1.0) [46%]	50%	666 (0.0) [1%]	+1329 (+2%)
Welfare organizations:											
- RSPCA	47,388 *	1.9	2665 (0.1) [6%]	31,105 (1.2) [66%]	70%	1033 (0.04) [2%]	34,803 (1.4) [74%]	11,740 (0.5) [25%]	26%	683 (0.0) [1%]	0 (0%)
- Other organizations	45,924 *	1.8	2460 (0.1) [5%]	29,222 (1.2) [65%]	69%	932 (0.04) [2%]	32,614 (1.3) [72%]	11,486 (0.5) [25%]	27%	970 (0.0) [2%]	+573 (+1%)
Sub-total:	93,312 *	3.7	5125 (0.2) [6%]	60,327 (2.4) [65%]	69%	1965 (0.08) [2%]	67,417 (2.7) [73%]	23,226 (0.9) [25%]	27%	1653 (0.1) [2%]	+573 (+1%)
Animal rescue groups	41,355 * (28,753 ^7^) *	1.7 (1.2 ^7^)	0 (0.0) [0%]	40,528 (1.6) [98%]	98%	0 (0.00) [0%]	40,528 (1.6) [98%]	827 (0.0) [2%]	2%	0 (0.0) [0%]	0 (0%)
Total intake	192,584 *	7.7									

^1^ Sums of intakes are identified with *; all other numbers are numbers of admissions; we defined an admission as commencing when a cat entered a municipal council pound, animal welfare shelter, or animal rescue group and ending only when one of four possible outcomes occurred: reclaimed by its owner, rehomed (i.e., allocated to a new owner), other outcome (e.g., died in the shelter, stolen, escaped etc.), or euthanized. During an admission, the cat might be transferred out to another organization (i.e., (another) municipal council, animal welfare organization, or animal rescue group) with the admission then ending with one of the four outcomes in a different organization from where the admission commenced. In contrast, we defined intake to a particular organization as all entries to that organization: (a) surrendered or stray cats brought in by the general public or authorities, and (b) cats transferred in from other organizations. ^2^ % are expressed of all admissions that ended in 2018–2019 or, for intake, % are expressed of all outcome events that occurred in year. ^3^ For admissions, cat was reclaimed or rehomed; for intakes, cat was reclaimed, rehomed or transferred out. ^4^ Outcome other than being reclaimed, rehomed, transferred or euthanized; includes cats that died in the shelter, were stolen or lost, etc. ^5^ Discrepancy for each organization was calculated (Appendix A: Table A3); a positive discrepancy indicates that the numbers available to experience an outcome (i.e., number in-care at start of year plus intake during year) were not all accounted for by numbers with outcomes and in-care at end of year; a negative discrepancy indicates that the numbers available to experience an outcome (i.e., number in-care at start of year plus intake during year) were less than (numbers with outcomes and in-care at end of year); discrepancy was calculated as the sum of discrepancies for each organization taking sign into account (i.e., sum of positive discrepancies minus sum of negative discrepancies). ^6^ % of (initial number in-care plus intake/admissions minus in care at end of year). ^7^ Estimated intake to animal rescue groups from general public.

**Table 3 animals-13-01771-t003:** Intakes/admissions and outcomes for cats entering municipal council pounds, welfare organizations and animal rescue groups in 2018–2019.

State/Territory and Organization	Intake/Admission for Year		Outcomes during Year								Discrepancy ^5^
			Returned to Owner (RTO)	Rehomed		Transferred Out	Released Alive ^3^	Euthanized		Other Outcomes ^4^	No. (%) ^6^
	No. ^1^	No. (Per 1000 Residents) [%] ^2^	No. (Per 1000 Residents) [%] ^2^	No. (Per 1000 Residents) [%] ^2^	% Rehomed of Unclaimed	No. (Per 1000 Residents) [%] ^2^	No. (Per 1000 Residents) [%] ^2^	No. (Per 1000 Residents) [%] ^2^	% Euthanized of Unclaimed	No. (Per 1000 Residents) [%] ^2^	
**NEW SOUTH WALES (human population 7,980,168)**											
Municipal councils that ran own pound service pooled	20,053 {19,928}	2.5 {2.5}	853 (0.1) [4%]	4326 (0.5) [22%]	23%	6259 (0.8) [31%]	11,438 (1.4) [57%]	8176 (1.0) [41%]	43%	295 (0.0) [1%]	0 (0%)
**Welfare organizations:**											
- RSPCA	14,676 ^7^	1.8	563 (0.1) [4%]	7652 (1.0) [53%]	55%	288 (0.0) [2%]	8503 (1.1) [59%]	5735 (0.7) [40%]	41%	211 (0.0) [1%]	0 (0%)
- Animal Welfare League NSW	1415	0.2	17 (0.0) [1%]	986 (0.1) [83%]	85%	0 (0.0) [0%]	1003 (0.1) [85%]	179 (0.0) [15%]	15%	0 (0.0) [0%]	0 (0%)
- Sydney Dogs and Cats Home	2266	0.3	74 (0.0) [3%]	1627 (0.2) [76%]	79%	0 (0.0) [0%]	1701 (0.2) [80%]	432 (0.1) [20%]	21%	0 (0.0) [0%]	+133 (+6%)
- Cat Protection Society (NSW)	862	0.1	3 (0.0) [0%]	837 (0.1) [98%]	98%	0 (0.0) [0%]	840 (0.1) [98%]	10 (0.0) [1%]	1%	4 (0.0) [0%]	−45 (−5%)
- Pets for Life	67	0.0	1 (0.0) [2%]	57 (0.0) [92%]	93%	0 (0.0) [0%]	58 (0.0) [94%]	1 (0.0) [2%]	2%	3 (0.0) [5%]	0 (0%)
Animal rescue groups pooled	6547 [NA ^8^]	0.8 [NA ^8^]	0 (0.0) [0%]	6416 (0.8) [98%]	98%	0 (0.0) [0%]	6416 (0.8) [98%]	131 (0.0) [2%]	2%	0 (0.0) [0%]	0 (0%)
Admissions/outcomes for state	39,339	4.9	1511 (0.2) [3%]	21,901 (2.7) [49%]	50%		29,960 (3.8) [66%]	14,664 (1.8) [32%]	34%	513 (0.1) [1%]	+88 (+0%)
**VICTORIA (human population 6,462,019)**											
Municipal councils that ran own pound service pooled	13,150 {12,749}	2.0 {2.0}	1243 (0.2) [10%]	5751 (0.9) [44%]	49%	330 (0.1) [3%]	7324 (1.1) [57%]	5631 (0.9) [43%]	48%	0 (0.0) [0%]	+195 (+1%)
**Welfare organizations:**											
- RSPCA	11,209^7^	1.7	730 (0.1) [7%]	7500 (1.2) [68%]	72%	391 (0.1) [4%]	8621 (1.3) [78%]	2369 (0.4) [21%]	23%	113 (0.0) [1%]	0 (0%)
- Animal Aid Victoria	2925	0.5	434 (0.1) [15%]	1711 (0.3) [59%]	69%	32 (0.0) [1%]	2177 (0.3) [75%]	692 (0.1) [24%]	28%	42 (0.0) [1%]	+14 (+0%)
- Lost Dogs Home	10,549	1.6	881 (0.1) [8%]	3867 (0.6) [37%]	40%	0 (0.0) [0%]	4748 (0.7) [46%]	5029 (0.8) [48%]	53%	654 (0.1) [6%]	+118 (+1%)
- Lort Smith Animal Hospital (Animal Welfare League)	938	0.1	0 (0.0) [0%]	522 (0.1) [75%]	75%	37 (0.0) [5%]	559 (0.1) [80%]	37 (0.0) [5%]	5%	100 (0.0) [11%]	+242 (+26%)
- Cat Protection Society of Victoria	1400	0.2	105 (0.0) [7%]	1274 (0.2) [84%]	90%	0 (0.0) [0%]	1379 (0.2) [90%]	146 (0.0) [10%]	10%	0 (0.0) [0%]	−125 (−9%)
- Geelong Animal Welfare Society	2981	0.5	203 (0.0) [7%]	2127 (0.3) [70%]	75%	366 (0.1) [12%]	2696 (0.4) [89%]	342 (0.1) [11%]	12%	0 (0.0) [0%]	−57 (−2%)
Animal rescue groups pooled	10,897 [10,107 ^8^]	1.7 [1.6 ^8^]	0 (0.0) [0%]	10,679 (1.7) [98%]	98%	0 (0.0) [0%]	10,679 (1.7) [98%]	218 (0.0) [2%]	2%	0 (0.0) [0%]	0 (0%)
Admissions/outcomes for state	52,893	8.2	3596 (0.6) [7%]	33,431 (5.2) [62%]	67%		38,183 (5.9) [71%]	14,464 (2.2) [27%]	29%	909 (0.1) [2%]	+387 (+1%)
**SOUTH AUSTRALIA (human population 1,736,527)**											
Municipal councils that ran own pound service pooled	2913 {1658}	1.7 {1.0}	60 (0.0) [3%]	1102 (0.6) [48%]	49%	0 (0.0) [0%]	1161 (0.7) [50%]	1017 (0.6) [44%]	45%	132 (0.1) [6%]	+601 (+21%)
**Welfare organizations:**											
- RSPCA	4693^7^	2.7	265 (0.2) [6%]	3317 (1.9) [70%]	74%	19 (0.0) [0%]	3601 (2.1) [75%]	1121 (0.6) [23%]	25%	50 (0.0) [1%]	0 (0%)
- Animal Welfare League SA	6781	3.9	142 (0.1) [2%]	3415 (2.0) [50%]	51%	51 (0.0) [1%]	3608 (2.1) [53%]	3120 (1.8) [46%]	47%	53 (0.0) [1%]	0 (0%)
Animal rescue groups pooled	4550 [4480 ^8^]	2.6 [2.6 ^8^]	0 (0.0) [0%]	4459 (2.6) [98%]	98%	0 (0.0) [0%]	4459 (2.6) [98%]	91 (0.1) [2%]	2%	0 (0.0) [0%]	0 (0%)
Admissions/outcomes for state	18,867	10.9	467 (0.3) [3%]	12,293 (7.1) [67%]	68%		12,829 (7.4) [70%]	5349 (3.1) [29%]	30%	235 (0.1) [1%]	+601 (+3%)
**AUSTRALIAN CAPITAL TERRITORY (human population 420,379)**											
**Welfare organizations:**											
- RSPCA	1683 ^7^	4.0	139 (0.3) [8%]	1186 (2.8) [70%]	77%	9 (0.0) [1%]	1334 (3.2) [79%]	318 (0.8) [19%]	21%	34 (0.1) [2%]	0 (0%)
Animal rescue groups pooled	808 [799 ^8^]	1.9 [1.9 ^8^]	0 (0.0) [0%]	792 (1.9) [98%]	98%	0 (0.0) [0%]	792 (1.9) [98%]	16 (0.0) [2%]	2%	0 (0.0) [0%]	0 (0%)
Admissions/outcomes for territory	2482	5.9	139 (0.3) [6%]	1978 (4.7) [79%]	84%		2126 (5.1) [85%]	334 (0.8) [13%]	14%	34 (0.1) [1%]	0 (0%)
**QUEENSLAND (human population 5,009,424)**											
Councils pound service pooled	11,446 {9570}	2.3 {1.9}	1182 (0.2) [11%]	1334 (0.3) [12%]	14%	2695 (0.5) [25%]	5063 (1.0) [46%]	5597 (1.1) [51%]	57%	148 (0.0) [1%]	+330 (+3%)
**Welfare organizations:**											
- RSPCA	12,621 ^7^	2.5	901 (0.2) [7%]	9833 (2.0) [77%]	82%	233 (0.0) [2%]	10,967 (2.2) [85%]	1666 (0.3) [13%]	14%	205 (0.0) [2%]	0 (0%)
- Animal Welfare League QLD	4320	0.9	362 (0.1) [8%]	3440 (0.7) [80%]	88%	65 (0.0) [2%]	3867 (0.8) [90%]	307 (0.1) [7%]	8%	100 (0.0) [2%]	0 (0%)
Animal rescue groups pooled	11,560 [8567 ^8^]	2.3 [1.7 ^8^]	0 (0.0) [0%]	11,329 (2.3) [98%]	98%	0 (0.0) [0%]	11,329 (2.3) [98%]	231 (0.0) [2%]	2%	0 (0.0) [0%]	0 (0%)
Admissions/outcomes for state	36,954	7.4	2445 (0.5) [6%]	25,936 (5.2) [65%]	70%		31,226 (6.2) [79%]	7801 (1.6) [20%]	21%	453 (0.1) [1%]	+330 (+1%)
**NORTHERN TERRITORY (human population 247,058)**											
Councils pound service pooled	645 {590}	2.6 {2.4}	16 (0.1) [3%]	0 (0.0) [0%]	0%	82 (0.3) [13%]	98 (0.4) [15%]	547 (2.2) [85%]	87%	0 (0.0) [0%]	0 (0%)
**Welfare organizations:**											
- RSPCA	387	1.6	6 (0.0) [2%]	330 (1.3) [91%]	93%	0 (0.0) [0%]	336 (1.4) [93%]	20 (0.1) [6%]	6%	6 (0.0) [2%]	0 (0%)
Animal rescue groups pooled	398 [316]	1.6 [1.3]	0 (0.0) [0%]	390 (1.6) [98%]	98%	0 (0.0) [0%]	0 (0.0) [0%]	8 (0.0) [2%]	2%	0 (0.0) [0%]	0 (0%)
Admissions/outcomes for territory	1348	5.5	22 (0.1) [2%]	720 (2.9) [51%]	52%		434 (1.8) [31%]	575 (2.3) [41%]	42%	6 (0.0) [0%]	0 (0%)
**WESTERN AUSTRALIA (human population 2,594,181)**											
Councils pound service pooled	9709 {3635}	3.7 {1.4}	652 (0.3) [7%]	2126 (0.8) [22%]	24%	1637 (0.6) [17%]	0 (0.0) [0%]	5000 (1.9) [53%]	56%	91 (0.0) [1%]	+203 (+2%)
**Welfare organizations:**											
- RSPCA	890 ^7^	0.3	22 (0.0) [3%]	570 (0.2) [65%]	67%	93 (0.0) [11%]	685 (18.7) [78%]	135 (0.1) [15%]	16%	57 (0.0) [6%]	0 (0%)
- Cat Haven	8919	3.4	160 (0.1) [2%]	7176 (2.8) [83%]	84%	381 (0.1) [4%]	7717 (211.0) [89%]	965 (0.4) [11%]	11%	0 (0.0) [0%]	+293 (+3%)
Animal rescue groups pooled	5594 [3483 ^8^]	2.2 [1.3 ^8^]	0 (0.0) [0%]	5482 (2.1) [98%]	98%	0 (0.0) [0%]	0 (0.0) [0%]	112 (0.0) [2%]	2%	0 (0.0) [0%]	0 (0%)
Admissions/outcomes for state	23,001	8.9	834 (0.3) [3%]	15,354 (5.9) [62%]	64%		8402 (229.8) [34%]	6212 (2.4) [25%]	26%	148 (0.1) [1%]	+496 (+2%)
**TASMANIA (human population 528,298)**											
Councils pound service pooled	0 {0}	0.0 {0.0}									
**Welfare organizations:**											
- RSPCA	1229^7^	2.3	39 (0.1) [3%]	717 (1.4) [63%]	65%	0 (0.0) [0%]	756 (1.4) [66%]	376 (0.7) [33%]	34%	7 (0.0) [1%]	0 (0%)
- Just Cats	1040	2.0	15 (0.0) [1%]	932 (1.8) [90%]	91%	0 (0.0) [0%]	947 (1.8) [91%]	79 (0.1) [8%]	8%	14 (0.0) [1%]	0 (0%)
- Ten Lives	1461	2.8	63 (0.1) [4%]	1251 (2.4) [86%]	89%	0 (0.0) [0%]	1314 (2.5) [90%]	147 (0.3) [10%]	11%	0 (0.0) [0%]	0 (0%)
Animal rescue groups pooled	1001 [1001 ^8^]	1.9 [1.9 ^8^]	0 (0.0) [0%]	981 (1.9) [98%]	98%	0 (0.0) [0%]	0 (0.0) [0%]	20 (0.0) [2%]	2%	0 (0.0) [0%]	0 (0%)
Admissions/outcomes for territory	4731	9.0	117 (0.2) [3%]	3881 (7.3) [84%]	86%		3017 (5.7) [65%]	622 (1.2) [13%]	14%	21 (0.0) [0%]	0 (0%)

^1^ Estimated total intake or number of admissions ((intake based on supplied data only) [Intake from general public]); we defined an admission as commencing when a cat entered a municipal council pound, animal welfare shelter, or animal rescue group and ending only when one of four possible outcomes occurred: reclaimed by its owner, rehomed (i.e., allocated to a new owner), other outcome (e.g., died in the shelter, stolen, escaped etc.), or euthanized. During an admission, the cat might be transferred out to another organization (i.e., (another) municipal council, animal welfare organization, or animal rescue group) with the admission then ending with one of the four outcomes in a different organization from where the admission commenced. In contrast, we defined intake to a particular organization as all entries to that organization: (a) surrendered or stray cats brought in by the general public or authorities, etc., and (b) cats transferred in from other organizations. ^2^ For intakes, % are expressed of all outcome events that occurred during year; for admissions, % are expressed of all admissions that ended in year. ^3^ For admissions, cat was reclaimed or rehomed; for intakes, cat was reclaimed, rehomed, or transferred out. ^4^ Outcome other than being reclaimed, rehomed, transferred, or euthanized; includes cats that died in the shelter, were stolen or lost, etc. ^5^ Discrepancy for each organization was calculated (Appendix A Table A3); a positive discrepancy indicates that the numbers available to experience an outcome (i.e., number in-care at start of year plus intake during year) were not all accounted for by numbers with outcomes and in-care at end of year; a negative discrepancy indicates that the numbers available to experience an outcome (i.e., number in-care at start of year plus intake during year) were less than (numbers with outcomes and in-care at end of year); for each state/territory, discrepancy was calculated as the sum of discrepancies for each organization taking sign into account (i.e., sum of positive discrepancies minus sum of negative discrepancies). ^6^ % of (initial number in-care plus intake/admissions). ^7^ Intakes reported by RSPCA for the year as “numbers received” included numbers in care at the start of the year; we calculated intakes by deducting numbers in care at the start of the year from numbers received as reported by RSPCA. ^8^ Intakes by rescue groups obtained from PetRescue were assumed to consist of transfers from other organizations and intake from the general public; the latter was estimated for each state/territory as intake by rescue groups minus sum of transfers out from other organizations in the same state/territory. For New South Wales, intake by rescue groups was recorded by PetRescue (2 rescue group’s data added from Justice4Max data) as 4700, but the number transferred out from municipal councils and RSCA was 6547. Thus, this estimate of intake by rescue groups appears to be an underestimate, and intake from the general public was not estimated.

**Table 4 animals-13-01771-t004:** Distributions of cat intakes and outcomes for municipal councils operating pounds in New South Wales and Victoria that had data available for 2018–2019 and had intakes >50 cats in 2018–19.

State and Statistic	Mean ^1^	Standard Deviation	Median	Lower Quartile Range (Minimum to 25th Percentile)	Upper Quartile Range (75th Percentile to Maximum)
**New South Wales (n = 61) ^2^**					
Human population (no. residents) in council area	70,591	96,182	8900	373,486	72,630 to 373,486
Intake	312	377	178	51 to 106	416 to 2437
Intake/1000 human residents in council area	21	39	7	0.2 to 2	24 to 245
Percentage reclaimed ^3^	5	4	4	0 to 2	5 to 18
Percentage rehomed ^3^	17	20	9	0 to 0	30 to 70
Percentage euthanized ^3^	46	27	42	2 to 25	67 to 100
Percentage of unclaimed rehomed ^3^	17	21	9	0 to 0	31 to 70
Percentage of unclaimed euthanized ^3^	48	27	46	2 to 27	67 to 100
**Victoria (n = 34)** ^4^					
Human population (no. residents) in council area	47,743	56,619	25,539	3862 to 11,521	62,585 to 255,367
Intake	389	414	217	54 to 125	476 to 1544
Intake/1000 human residents in council area	11	7	9	1 to 6	16 to 27
Percentage reclaimed ^3^	10	6	9	2 to 5	14 to 24
Percentage rehomed ^3^	37	24	37	0 to 15	58 to 76
Percentage euthanized ^3^	49	26	47	0 to 28	73 to 98
Percentage of unclaimed rehomed ^3^	42	27	41	0 to 16	65 to 86
Percentage of unclaimed euthanized ^3^	54	28	52	0 to 30	79 to 100

^1^ Unweighted means of intakes and means of percentages for each included council. ^2^ Of the 128 municipal councils in New South Wales, 111 operated pounds, 107 of those supplied data, and 61 of those had intakes >50. ^3^ % of all outcome events that occurred during year. ^4^ Of the 79 municipal councils in Victoria, 38 operated pounds, 37 of those supplied data, and 34 of those had intakes >50 cats.

**Table 5 animals-13-01771-t005:** Cat intakes and numbers euthanized for municipal councils, RSPCA, and Animal Welfare League in New South Wales by year from 2016–2017 to 2018–2019.

Organization	Year	Intake	% Change	No. of Cats Euthanized	% Change
Municipal councils pooled ^1^	2016/17	20,025/17,566 ^1^	+3%/+8% ^1, 2^	8104/6852	+4%/+13% ^1, 2^
	2017/18	20,630/19,044 ^1^	+13%/+5% ^1, 2^	8404/7757	+12%/+4% ^1, 2^
	2018/19	23,368/19,967 ^1^	+17%/+14% ^1, 3^	9419/8046	+16%/+17% ^1, 3^
RSPCA	2016/17	15,608	−2% ^2^	6571	−7%^2^
	2017/18	15,309	−4% ^2^	6118	−6%^2^
	2018/19	14,676	−6% ^3^	5735	−13%^3^
Animal Welfare League	2016/17	2236 ^4^	−28% ^2^	267	−20%^2^
	2017/18	1614 ^4^	−7% ^2^	214	−16%^2^
	2018/19	1506 ^4^	−33% ^3^	179	−33%^3^

^1^ First result is for all councils with data in the respective year (including intakes for councils that had contracted pound services to an animal welfare organization); second result is for the 97 of those councils where data were available for all 3 years; 2018/19 numbers differ from those in Table 3 as results in Table 3 are councils that operated pounds and had not contracted pound services to an animal welfare organization, and included imputed data for councils known to operate pounds but where data were not available. ^2^ Change relative to previous year expressed as percentage of number in previous year. ^3^ Change from 2016/17 to 2018/19 expressed as percentage of number in 2016/17. ^4^ Includes numbers in care at start of year; in 2018/19, 91 cats were in care at the start of the year and intake during that year was 1415.

## Data Availability

Raw data can be provided on request.

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
