# Peer review of "Stray and Owner-Relinquished Cats in Australia—Estimation of Numbers Entering Municipal Pounds, Shelters and Rescue Groups and Their Outcomes"

_animals, 2023, doi:10.3390/ani13111771_

Round 1

Reviewer 1 Report

See attached. 

Author Response

Attached response to reviewer 1 as a file

Reviewer 2 Report

The following study aimed to develop a methodology to accurately assess the number of cats entering municipal pounds, animal welfare shelters, and rescue groups in Australia and report the outcomes of those admissions. The study also suggests this data will establish a baseline to assess future management strategies' effectiveness. Information regarding reclaimed, rehoming, and euthanasia of cats is an essential first step for any program aiming to reduce euthanasia numbers. However, it is unclear how collecting this descriptive data alone will allow the authors, or other researchers, to improve cat management in the future. In addition to collecting descriptive data, it is unclear if there are other research questions the authors plan to test (here or in another publication) or predictions on what they expected to find before collecting the data (hypothesis). For example,  comparison among areas (rural versus more urban) or differences between municipal pounds and welfare shelters or rescue groups could easily be explored with the data provided. Are there plans to pursue this? Besides a large data dump, it is unclear what the authors are trying to answer and how data collection will help. Having said this, I do think this data is valuable and deserves to be published after revisions.

Specific Comments:

1.      The authors state that collecting data at a national level would allow for the targeting and tracking of effective strategies, but we don't know who would use this info and how it would be done.

2.      Although accurate data is the first step of what is in place is often the first step, it is unclear how this will be applied. How does knowing how many cats and where they came from benefit humans, cats, and wildlife?

3.      The methods section is exceedingly long. I realize that this study's primary goal is to develop a methodology that future studies can use but is excessive. Tell the reader what they need to know and direct them to further information in an appendix or supplemental material.

4.      I am also concerned about the mechanism for counting initial entries vs. repeaters. I'm not sure this would be accurate unless they are all microchipped or tagged in some other way.

5.      Results- 306 municipal councils plus 43 animal welfare does not equal the 357 reported possible sources. Where are the others?

6.      Table 1 could be put in supplementary materials. Table 2 and 3, is there repeated info?

7.      Figure 1 a-d provides the same info as Figure 2. Figure 2 is sufficient. Likewise, Figure 3b is enough; no need for Figure 3a. I am unsure how important Table 4 is and if it can also be in supplementary materials.

8.      The authors recommend that statistics from shelter and rescue groups be publicly available because it would inform management strategies. Yet, they don't tell us how this would happen. Knowing numbers alone isn't very useful. What are the strategies for targeting high numbers (spay/neuter, behavioral assistance, farm cats, etc.)? What will they do to address the number of cats? What strategies are there?

9.      The manuscript is way too long and unnecessarily so. The authors repeat statements frequently, and it does not add value. The discussion is far too long and repetitive.

Author Response

Attached response to reviewer 2 as a file
